# CHARM: Calibrating Reward Models With Chatbot Arena Scores

## Abstract

Reward models (RMs) play a crucial role in Reinforcement Learning from Human Feedback by serving as proxies for human preferences in aligning large language models. However, they suffer from various biases which could lead to reward hacking. In this paper, we identify a model preference bias in RMs, where they systematically assign disproportionately high scores to responses from certain policy models, leading to unfair judgments. To mitigate this bias, we propose a calibration method named **CH**atbot **A**rena calibrated **R**eward **M**odeling (**CHARM**) that leverages Elo scores from the Chatbot Arena to construct debiased preference datasets and adjust reward model scoring. We conduct extensive experiments on reward model benchmarks and human preference alignment. Results demonstrate that our calibrated RMs achieve improved evaluation accuracy on RM-Bench and the Chat-Hard domain of RewardBench and exhibit a stronger correlation with human preferences by producing scores more closely aligned with Elo rankings. Beyond this, **CHARM** enhances robustness to stylistic variations, mitigates implicit pattern bias, and generalizes to unseen models. These results demonstrate that **CHARM** provides a simple, effective, and broadly applicable approach to building more reliable and fair reward models.

## 1 Introduction

Reinforcement Learning from Human Feedback (RLHF; Ouyang et al., 2022; Christiano et al., 2017) has emerged as a fundamental approach for aligning large language models (LLMs) with human values, ensuring they generate helpful, coherent, and safe responses (Achiam et al., 2023; Touvron et al., 2023; Gemini Team et al., 2023; Bai et al., 2023). At the core of RLHF are reward models (RMs). RMs are typically trained on pairwise preference data, where human annotators evaluate multiple model-generated responses and rank them based on specific criteria (Ouyang et al., 2022; Lee et al., 2024). Given these ranked preferences, the RM learns to predict which responses humans would favor, effectively acting as an automated judge in place of human raters.

However, reward models are not infallible; their inherent biases can compromise the fairness of evaluations and allow policy models to exploit these vulnerabilities through reward hacking (Skalse et al., 2022). In such scenarios, language models optimize their outputs to maximize the reward, often in ways that deviate from genuine human preferences (Eisenstein et al., 2023; Gao et al., 2023; Pang et al., 2022). Recent studies have revealed several such biases, which often manifest as a preference for specific content patterns, such as length bias (Huang et al., 2025) or style bias (Zhang et al., 2024b). The existence of these biases means a model can achieve a high score simply by generating longer or more elaborately styled text rather than by genuinely improving the reliability of its content, potentially leading to deceptive or unintended outcomes.

In this paper, by analyzing the score distributions of various reward models across a wide range of policy models, we identify a different and more subtle type of bias. We term this **Model Preference Bias**, as it manifests in reward models systematically giving disproportionately high scores to certain policy models, beyond what is justified by human preferences. We propose a calibration method named **CH**atbot **A**rena calibrated **R**eward **M**odeling (**CHARM**), which utilizes these over-valued policy models as generators of false positives and constructs debiased preference pairs by leveraging Elo scores from Chatbot Arena (Chiang et al., 2024). Through extensive experiments, we demonstrate

that RMs calibrated with **CHARM** not only achieve better performance on reward model benchmarks but are also more aligned with genuine human preferences. Furthermore, our analysis reveals that the calibrated models are more robust to stylistic changes in responses; our method adjusts the score distribution at a model level, which can be understood as a form of implicit pattern calibration.

## 2 RELATED WORK

**Reward Models** A reward model assigns scores to responses generated by large language models, helping rank and select the most human-aligned outputs (Ouyang et al., 2022). Formally, let $\mathcal{D} = \{(x, y)\}$ represent a dataset of instruction-response pairs, where $x \in \mathcal{X}$ is an instruction and $y \in \mathcal{Y}$ is a response. A reward model $r_\phi : \mathcal{X} \times \mathcal{Y} \to \mathbb{R}$ predicts the score $r_\phi(x, y)$ for a response $y$ conditioned on an instruction $x$. Most RMs are trained using pairwise preference data, which consists of triplets $(x, y^+, y^-)$, where $y^+$ is the preferred response over $y^-$. The RM is trained to optimize a Bradley-Terry pairwise ranking loss:

$$\mathcal{L}(\phi) = -\mathbb{E}_{(x,y^+,y^-)\sim\mathcal{D}}\Big[ \log \sigma\big(r_\phi(x, y^+) - r_\phi(x, y^-)\big)\Big] \tag{1}$$

**LLM Evaluation** Both automated benchmarks and human evaluation are widely used in LLM evaluation. A series of benchmarks such as MT-Bench (Zheng et al., 2023b), Alpaca-Eval (Dubois et al., 2023), and Arena-hard (Li et al., 2024) employ LLM-as-a-judge systems to assess the quality of model responses on a fixed set of prompts. A series of works, such as Ultrafeedback (Cui et al., 2024) and RLAIF (Lee et al., 2024), use LLMs for preference annotation which correlate training signals with evaluation measures. On the other side, ChatBot Arena (Chiang et al., 2024) employs a crowdsourced, pairwise comparison system where users challenge two anonymous models with prompts and select their preferred response. This vast collection of human preference data is then aggregated to compute a dynamic Elo rating for each model, which serves as a widely recognized proxy for genuine human preference.

**Bias Mitigation** Reward models and LLM-as-a-judge might bring biases and affect the accuracy of the evaluation. Park et al. (2024) identified six distinct types of bias in evaluation models and leveraged LLMs to construct a debiased dataset. Beyond these biases, Li et al. (2025) found that judge models may develop bias, favoring content generated by themselves or closely related LLMs due to their exposure to synthetic data. Dubois et al. (2024) proposed a regression-based method to mitigate length bias, while Huang et al. (2025) introduced a post hoc calibration technique for reward models.

**Classifier Calibration** In the context of supervised learning, a K-class probabilistic classifier is considered well-calibrated if, among test instances receiving a predicted K-dimensional probability vector **s**, the empirical class distribution is approximately consistent with **s** (Silva Filho et al., 2023). For reward models, calibration refers to debiasing the model so that its outputs better align with human preferences or other certain metrics. This notion of calibration operates at the model- or class-level, rather than at the sample-wise level as in standard classifier calibration.

## 3 MODEL PREFERENCE BIAS: AN EMPIRICAL STUDY

Reward models are designed with the primary goal of aligning model responses with human preferences. One of the most direct reflections of human preferences in large-scale AI evaluation is Chatbot Arena (Zheng et al., 2023b), where real users interact with language models and rank them based on their responses. Given that Chatbot Arena represents a universal distribution of real-world prompts, we can make the assumption that an ideal RM should produce scores that strongly correlate with the platform's Elo ratings.

To validate this, we use AlpacaEval (Li et al., 2023) as our evaluation set, which consists of 805 carefully curated questions. This dataset has been shown to exhibit a 98% Spearman correlation with Chatbot Arena. We evaluated five popular RMs (Liu et al., 2024; Dong et al., 2023; Xiong et al., 2024; Yang et al., 2024) on a diverse set of policy models with varying Elo scores (Since some models on AlpacaEval did not participate in the Chatbot Arena, their Elo scores are unavailable. More results

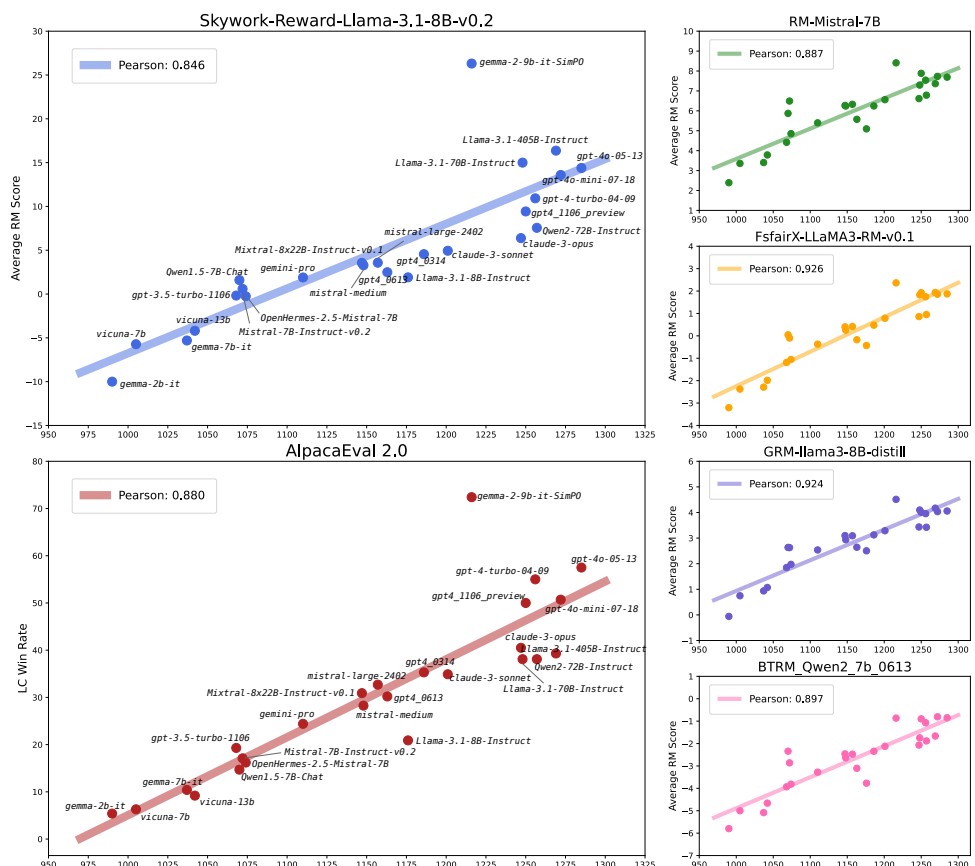

Figure 1: Average reward model scores across policy models on AlpacaEval. The x-axis represents arena Elo scores. The left lower plot illustrates the Length-Controlled win rates of these models on AlpacaEval.

and statistics are in Appendix A.1 and Appendix A.2). For each policy model, we calculated the average score from each RM. We also display the Length-Controlled win rates of these policy models on AlpacaEval (Dubois et al., 2024). The results are shown in Figure 1:

**Observation 1: RM Scores Correlate Positively with Human Preferences**    From Figure 1, we observe that models with higher Elo scores in Chatbot Arena tend to receive higher RM scores on their responses. This supports our initial assumption that an ideal RM should reflect human preference rankings. We compute the Pearson correlation between policy models' RM and Elo scores for each RM (Also see Figure 1). The results indicate a strong positive correlation across all tested RMs.

**Observation 2: RMs May Favor Certain Policy Models Unfairly**    Although RM scores exhibit an overall alignment with human preferences, they sometimes deviate for specific policy models, assigning scores that are inconsistent with the models' Elo scores. Some policy models, such as Gemma-2-9b-it-SimPO (Meng et al., 2024), receive disproportionately high RM scores, sometimes even surpassing significantly stronger models based on Elo rankings. A similar trend is observed in the AlpacaEval leaderboard, suggesting that this bias may also exist in LLM-as-a-judge systems. Notably, the over-valued models share a similarity in that they undergo preference optimization, implying a potential source of systematic biases.

Based on these observations, we define this issue as **Model Preference Bias**, which occurs when reward models systematically assign unjustifiably high scores to certain policy models. This overestimation makes RMs fail to provide fair evaluations. To address this bias, we detail our proposed calibration method in the next section. More analysis of model preference bias can be found in Section 6.

# 4 METHODOLOGY

## 4.1 CHARM: CHATBOT ARENA CALIBRATED REWARD MODELING

Given a set of instructions $\mathcal{X} = \{x_i\}_{i=1}^N$ and two policy models, one over-valued model $\pi_O$ and one reference model $\pi_R$, for each instruction $x_i$, the two models generate responses $y_i^O \sim \pi_O$ and $y_i^R \sim \pi_R$. The reward model $r_\phi$ assigns scores to these responses, producing $s_i^O = r_\phi(x_i, y_i^O)$ and $s_i^R = r_\phi(x_i, y_i^R)$, resulting in a preference dataset $\mathcal{D} = \{(x_i, y_i^+, y_i^-) \mid y_i^+ = \arg\max(s_i^O, s_i^R)\}_{i=1}^N$. Since the reward model $r_\phi$ overestimates the responses from model $\pi_O$, the resulting preference dataset $\mathcal{D}$ inherits this bias. To address this issue, **CHARM** reconstructs a debiased preference dataset to mitigate the preference bias in reward modeling.

The Elo rating system (Elo, 1967) provides a probabilistic model for ranking players (or models, in this case) based on their relative performance. Following Chatbot Arena's implementation where a model gets a score of 1 for a win, 0.5 for a tie, and 0 for a loss, for an over-valued model with $\text{Elo}_O$ and a reference model with $\text{Elo}_R$, the expected win rate of the over-valued model $\mathbb{P}(O)$ is defined as Equation 2. $\mathbb{P}(O)$ can also be expressed as a weighted sum of the probabilities of the over-valued model getting a win $\mathbb{P}_{win}$ and a tie $\mathbb{P}_{tie}$:

$$\mathbb{P}(O) = \frac{1}{1 + 10^{(\text{Elo}_R - \text{Elo}_O)/400}} = \mathbb{P}_{win} + 0.5\mathbb{P}_{tie} \qquad (2)$$

Ties in RM are rare unless both models produce identical responses, naturally requiring $\mathbb{P}_{tie} \to 0$. Nonetheless, Chatbot Arena's scoring implementation (1/0.5/0 for win/tie/loss) allows us to evenly split $\mathbb{P}_{tie}$ between wins and losses, maintaining equivalent Elo scores. Therefore, our Elo-derived win rate can be directly applicable to a strict win/loss scenario for RMs. Given the observation in section 3 that RM scores are correlated with Elo scores, we contend that if the RM were perfectly aligned with human preferences, its empirical win rate should match this probability:

$$\hat{\mathbb{P}}(O) = \frac{1}{N} \sum_{i=1}^N \sigma(s_i^O - s_i^R) \approx \mathbb{P}(O) \qquad (3)$$

However, in practice, we find that there exist deviations between $\hat{\mathbb{P}}(O)$ and $\mathbb{P}(O)$ because of model preference bias. To correct this bias, we seek a transformation of RM scores such that the empirical win rate $\hat{\mathbb{P}}'(O)$ after calibration better aligns with the expected win probability $\mathbb{P}(O)$. We introduce a score offset $\Delta$ applied to the RM scores of over-valued policy model's responses: $s_i'^O = s_i^O + \Delta$, then the calibrated empirical win rate will be: $\hat{\mathbb{P}}'(O) = \frac{1}{N} \sum_{i=1}^N \sigma(s_i'^O - s_i^R)$. Our goal is to find a $\Delta$ that minimizes the deviation from the theoretical probability. We optimize $\Delta$ by minimizing the MSE loss:

$$\mathcal{L}(\Delta) = \mathbf{MSE}\Big( \frac{1}{N} \sum_{i=1}^N \sigma(s_i^O + \Delta - s_i^R), \mathbb{P}(O) \Big) \qquad (4)$$

After determining the offset $\Delta$, we can construct a calibrated preference dataset $\mathcal{D}' = \{(x_i, y_i^+, y_i^-) \mid y_i^+ = \arg\max(s_i^O + \Delta, s_i^R)\}_{i=1}^N$ for further reward model training.

## 4.2 A METRIC FOR MODEL PREFERENCE BIAS MEASUREMENT

To quantify the misalignment between a reward model's preference for different policy models and human preferences, we introduce a Mismatch Degree metric. This metric shows the discrepancy between the reward model's scoring and the expected human preference reflected by Elo scores, measuring the degree of RM's model preference bias.

Given a model $\pi_O$, a reference model $\pi_R$, and a preference dataset built upon them, we define the Mismatch Degree (MD) between them as:

$$\mathbf{MD}(\pi_O, \pi_R) = \left| \frac{\hat{\mathbb{P}}(O) - \mathbb{P}(O)}{\max(\mathbb{P}(O), 1 - \mathbb{P}(O))} \right| \qquad (5)$$

where $\hat{\mathbb{P}}(O)$ is the probability of model $\pi_O$ winning against $\pi_R$ according to the reward model's scores. $\mathbb{P}(O)$ is the expected win rate of $\pi_O$ over $\pi_R$, derived from their Elo scores in Chatbot Arena. This metric captures how much the reward model's judgments deviate from the expected human preference. A positive $\hat{\mathbb{P}}(O) - \mathbb{P}(O)$ indicates that the reward model over-values model $\pi_O$ relative to what is expected from human preferences while a negative value indicates an under-value.

## 5 EXPERIMENTS

In this section, we aim to address the following questions through experiments to validate the effectiveness of **CHARM**:

**Question 1** Does **CHARM** enhance the reward model's judging capability, leading to more accurate and reliable evaluations?

• We evaluate calibrated reward models on benchmarks such as RM-Bench and RewardBench, which consist of diverse instructions paired with two candidate responses. The reward model must assess and select the better response, providing a robust framework to measure its judging capability.

**Question 2** Does **CHARM** successfully reduce model preference bias, improving alignment with human preferences?

• We construct a battlefield using responses from various LLMs on AlpacaEval. Each response is scored by the reward models, allowing us to compute pairwise win rates between models. We then compare these RM-derived win rates against Elo-derived win rates obtained from Chatbot Arena, which reflect human preferences. By analyzing their alignment, we assess whether the calibration process effectively reduces bias and produces rankings that more accurately reflect human judgments.

**Question 3** Does **CHARM** boost model preference tuning, resulting in a better alignment of the post-trained model?

• We trained a DPO model using the CHARM-calibrated reward model. By analyzing the downstream performance of the resulting policy model, we directly evaluate the effectiveness of CHARM in improving post-training alignment.

### 5.1 IMPLEMENTATION DETAILS

For preference dataset construction, we use Preference700K (Dong et al., 2024), a comprehensive dataset that aggregates preference data from eight sources. We randomly sampled 20K instructions from Preference700K and generated corresponding responses using selected over-valued and reference models. These responses were then scored by a reward model that we later calibrate. This process produced the uncalibrated preference dataset, which served as the foundation for applying **CHARM** to construct the calibrated preference dataset.

We set temperature $\tau = 0.7$ and Top_p = 0.9 during inference. We selected five reward models for calibration: *Skywork-Reward-Llama-3.1-8B-v0.2* (Liu et al., 2024), *RM-Mistral-7B*, *FsfairX-LLaMA3-RM-v0.1* (Dong et al., 2023; Xiong et al., 2024), *GRM-llama3-8B-distill* (Yang et al., 2024), and *BTRM-Qwen2-7b-0613*. During reward model fine-tuning, we used the Adam optimizer (Kingma & Ba, 2014) with a learning rate of 2e-6, a weight decay of 0.001, and a cosine learning rate scheduler. The models were trained for 1 epoch.

### 5.2 EXPERIMENT RESULTS

#### 5.2.1 RESULTS ON REWARD MODEL BENCHMARKS

After obtaining reward scores for various policy models on AlpacaEval, as described in Section 3, we derive a scoring profile for each model. We then select a strong model *GPT-4o-mini* as our reference and compute the Mismatch Degree of every other policy model relative to it. Finally, we choose the model pair with the highest Mismatch Degree, *gemma-2-9b-it-SimPO* vs. *GPT-4o-mini*, for our further calibration.

| Reward Models | Mismatch Degree | RM-Bench | | | | | | | | RewardBench |
|---|---|---|---|---|---|---|---|---|---|---|
| | | Chat | Math | Code | Safety | Hard | Normal | Easy | *Avg* | Chat-Hard |
| Skywork-RM | | 68.7 | 62.0 | 52.8 | 95.9 | 47.5 | 73.7 | 88.4 | 69.9 | 88.8 |
| *w/o calibration* | 0.639 | 68.9 | 61.9 | 53.1 | 95.9 | 47.6 | 73.8 | 88.5 | 70.0 | 88.8 |
| *w/ calibration* | | **73.9** | 62.4 | 53.9 | 95.8 | 49.3 | 75.8 | 89.4 | 71.5 | **89.4** |
| FsfairX-RM | | 62.5 | 63.2 | 54.6 | 90.4 | 44.9 | 71.6 | 86.5 | 67.7 | 65.3 |
| *w/o calibration* | 0.554 | 63.1 | 63.4 | 53.6 | 90.4 | 45.8 | 71.6 | 85.4 | 67.6 | 65.1 |
| *w/ calibration* | | **64.5** | 63.3 | 56.0 | 90.0 | 45.4 | 72.6 | 87.5 | 68.5 | **65.7** |
| Mistral-RM | | 60.8 | 56.6 | 52.6 | 88.7 | 37.5 | 68.2 | 88.3 | 64.7 | 60.5 |
| *w/o calibration* | 0.528 | 61.4 | 57.4 | 53.0 | 88.9 | 40.0 | 68.8 | 86.7 | 65.2 | 62.5 |
| *w/ calibration* | | **63.2** | 57.0 | 52.4 | 88.3 | 36.3 | 69.8 | 89.5 | 65.2 | **65.1** |
| GRM-RM | | 63.6 | 62.0 | 56.9 | 89.1 | 49.6 | 71.8 | 82.2 | 67.9 | 68.4 |
| *w/o calibration* | 0.508 | 63.6 | 62.4 | 58.3 | 89.5 | 49.8 | 72.7 | 82.9 | 68.4 | 68.8 |
| *w/ calibration* | | **66.2** | 62.6 | 58.0 | 89.3 | 48.3 | 73.9 | 84.9 | 69.0 | **68.9** |
| BTRM-RM | | 60.0 | 61.3 | 53.8 | 89.9 | 37.1 | 71.0 | 90.7 | 66.3 | 58.1 |
| *w/o calibration* | 0.162 | 58.5 | 61.5 | 54.1 | 89.1 | 35.3 | 70.8 | 91.3 | 65.8 | 58.7 |
| *w/ calibration* | | 60.2 | 60.5 | 53.8 | 89.6 | 34.8 | 71.1 | 92.1 | 66.0 | 57.8 |

Table 1: Results of different RM versions on the RM-Bench and RewardBench benchmark. We highlight the performance gains on RM-Bench Chat and RewardBench Chat-Hard.

We choose five reward models from the RM-Bench leaderboard, each exhibiting varying levels of performance. We compute their MD on the selected model pair, revealing distinct deviations in how they value *Gemma-2-9b-it-SimPO* relative to human preferences. These reward models serve as the base models for our experiments. Following the methodology described in Section 5.1, we construct both uncalibrated and calibrated preference datasets for each reward model.

We evaluated three versions of each reward model on the benchmark: (**1**) the original reward model, (**2**) the reward model trained on the uncalibrated dataset, and (**3**) the reward model trained on the calibrated dataset. We select RM-Bench and RewardBench as our test benchmarks. For RewardBench, only the more challenging Chat-Hard domain is reported since other domains have shown nearly saturated results. The overall results are displayed in Table 1.

From the benchmark results across different versions of the reward model, we can summarize the following findings:

**Finding 1: Uncalibrated Preference Datasets Lead to Minimal Performance Gains**   Across all evaluated models, uncalibrated training led to minimal or no improvement over the original reward model. While an uncalibrated preference dataset introduces additional preference data, it does not explicitly correct biases. The underlying issues in the RM's decision boundaries remain unaddressed, resulting in no meaningful shift in performance.

**Finding 2: CHARM Enhances Overall Performance, Especially in Chat Domain**   Training on the calibrated preference dataset enhances reward model performance across benchmarks. On average, RM-Bench scores improved by +0.74 points, with Skywork-RM showing the largest gain of +1.6 points. Among all evaluated tasks, Chat performance saw the most substantial improvement after calibration. Skywork-RM achieved the largest gain of +5.2 points, followed by Mistral-RM of +2.4 points and FsfairX-RM of +2.0 points. And a similar trend was observed in RewardBench Chat-Hard.

### 5.2.2   RESULTS ON HUMAN PREFERENCE ALIGNMENT

One of the primary objectives of our calibration method is to better align the reward model's judgment with human preferences. To evaluate whether **CHARM** effectively mitigates model preference bias, we designed experiments on the AlpacaEval dataset, which contains a wide range of policy models and their responses, making it convenient for constructing pairwise battles. We selected 24 policy models and their responses from AlpacaEval and then used both the original and calibrated Skywork-RM to score these responses. After scoring, we built pairwise comparisons from the RM-assigned

| Models | IFEval | | | |
|---|---|---|---|---|
| | Inst. Loose | Inst. Strict | Prompt Loose | Prompt Strict |
| Qwen2.5-7B-Instruct | 72.06 | 67.51 | 62.11 | 56.75 |
| DPO (Original RM) | 72.30 | 67.87 | 62.29 | 56.19 |
| DPO (CHARM-Calibrated RM) | 74.46 | 68.47 | 64.88 | 57.67 |

Table 2: Performance of Qwen2.5-7B-Instruct on IFEval under different DPO settings.

scores to obtain RM-derived win rates between model pairs. These win rates were compared against Elo-derived win rates, allowing us to compute the Mismatch Degree as a measure of model preference bias. The results are presented in Figure 2.

**Finding 3: CHARM Reduces Model Preference Bias** From the win rate comparison, we observe that models' performance against *GPT-4o-mini-2024-07-18* and *Gemma-2-9b-it-SimPO* exhibits stronger alignment with ideal human preferences. Specifically, the win rates derived from RM scores are now closer to those based on Elo scores. Additionally, we compute the MD across different models, and the results reveal a clear reduction in MD. This indicates that calibration effectively mitigates model preference bias for both the over-valued and reference models, demonstrating improved alignment of calibrated reward models with human preferences.

**Finding 4: CHARM Generalizes to Unseen Models** We further selected *Qwen2-72B-Instruct* and *mistral-large-2402*, two policy models that were not used during the calibration process, to assess whether our method generalizes to unseen LLMs. Results in Figure 2 (right) show that the calibrated reward model maintains a stronger correlation with human preferences even on these unseen models. Notably, this generalization emerges despite using only two policy models during calibration, suggesting the existence of more inherent factors underlying model preference bias. We make further analysis in the following section.

### 5.2.3 RESULTS ON POST TRAINING ALIGNMENT

To determine whether CHARM offers practical post-training alignment benefits beyond improvements on RM benchmarks, we evaluate its impact in a real post-training setting. Specifically, we trained a DPO model using the CHARM-calibrated reward model. Since model preference bias may originate from biased preference-learning datasets as discussed in the Appendix A.1, offline DPO provides an ideal controlled environment for assessing the effectiveness of CHARM.

We generated responses from *GPT-4o-mini* and *Gemma-2-9b-it-SimPO* on Preference20K and obtained reward scores from both the original Skywork-RM and the CHARM-calibrated Skywork-RM. Using these scores, we constructed two DPO datasets and trained *Qwen2.5-7B-Instruct* separately on each. The resulting models were evaluated on IFEval, and the results are shown in Table 2.

**Finding 5: CHARM Improves Downstream Model Performance** Models trained with CHARM-calibrated RM consistently outperform those trained with the original RM and the baseline model across all IFEval metrics. Notably, the uncalibrated RM provides marginal gains or even degrades performance, while CHARM yields consistent improvements. These results demonstrate that correcting model preference bias is beneficial for enhancing downstream model performance.

## 6 ANALYSIS

### 6.1 MISMATCH DEGREE SERVES AS AN INDICATOR OF CALIBRATION NEED

We observe a potential correlation between Mismatch Degree and performance improvement after calibration in Table 1. Notably, Skywork-RM, which exhibited the highest MD, achieved the most significant performance gains. In contrast, BTRM, which had the lowest MD, even experienced a slight performance degradation. To further investigate the relationship between MD and calibration

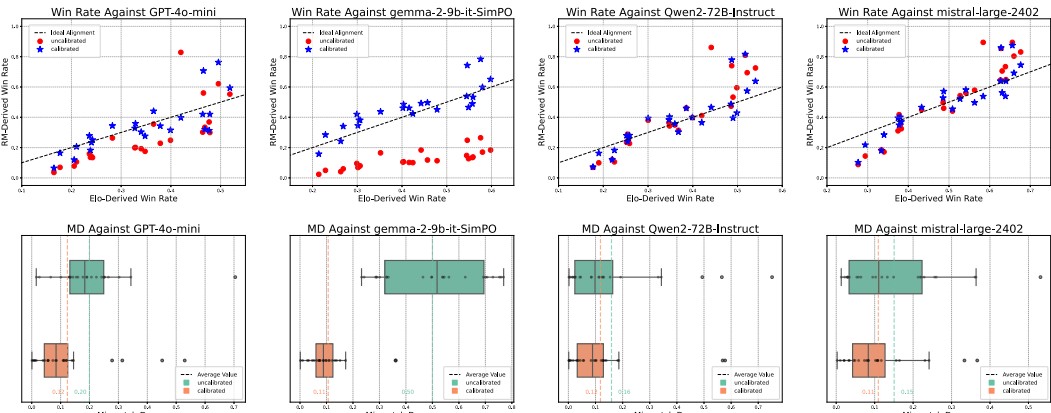

Figure 2: Win rates and Mismatch Degrees before and after calibration. In the win rate plots, the x-axis is the expected win rates calculated based on the models' Elo scores, while the y-axis is the win rates derived from the reward model scores. Points closer to the dotted line indicate a better alignment between the reward model and human preferences.

| Over-valued Models | Ref Models | Mismatch Degree | RM-Bench | | | | |
|---|---|---|---|---|---|---|---|
| | | | Chat | Math | Code | Safety | *Avg* |
| **Skywork-RM** | | | 68.7 | 62.0 | 52.8 | 95.9 | 69.9 |
| *gemma-2-9b-it-SimPO* | *GPT-4o-mini* | 0.639 | 73.9 | 62.4 | 53.9 | 95.8 | 71.5 |
| *Gemma-2-27b-it* | *GPT-4o-mini* | 0.225 | 70.7 | 61.9 | 52.9 | 96.7 | 70.5 |
| *Gemma-2-9b-it* | *GPT-4o-mini* | 0.155 | 70.9 | 62.2 | 52.9 | 96.6 | 70.7 |
| *Qwen2.5-72B-Instruct* | *GPT-4o-mini* | 0.088 | 70.5 | 62.1 | 52.6 | 96.2 | 70.4 |
| *Llama-3.1-70B-Instruct* | *GPT-4o-mini* | 0.048 | 68.9 | 61.9 | 53.0 | 96.0 | 70.0 |
| *Llama-3.1-8B-Instruct* | *GPT-4o-mini* | 0.032 | 68.8 | 61.6 | 52.5 | 96.2 | 69.8 |
| *gemma-2-9b-it-SimPO* | *Gemma-2-9b-it* | 0.582 | 70.7 | 63.2 | 54.6 | 94.7 | 70.8 |
| *gemma-2-9b-it-SimPO* | *Gemma-2-27b-it* | 0.633 | 70.0 | 63.1 | 54.0 | 94.7 | 70.4 |
| *gemma-2-9b-it-SimPO* | *Llama-3.1-8B-Instruct* | 0.506 | 71.4 | 64.5 | 53.7 | 95.9 | 71.4 |
| *gemma-2-9b-it-SimPO* | *Llama-3.1-70B-Instruct* | 0.675 | 72.2 | 64.7 | 55.0 | 95.2 | 71.8 |
| *gemma-2-9b-it-SimPO* | *Qwen2.5-72B-Instruct* | 0.625 | 70.9 | 62.4 | 53.8 | 96.0 | 70.8 |

Table 3: Impact of Mismatch Degree on calibration effectiveness. We construct different model pairs with varing MD and perform CHARM on them.

effectiveness, we design additional experiments to analyze how MD influences the impact of reward model calibration. We fix Skywork-RM as the reward model and construct different model pairs to produce varying levels of mismatch degree. We then repeat the same experiments on each model pair. The results are presented in Table 3.

By analyzing the results, we observe that MD serves as a strong indicator of a model's misalignment and the potential benefits of calibration. Models with higher MD values tend to exhibit greater improvements after calibration. For instance, *gemma-2-9b-it-SimPO* vs. *GPT-4o-mini* (MD = 0.639) and *gemma-2-9b-it-SimPO* vs. *Llama-3.1-70B-Instruct* (MD = 0.675) benefit the most from calibration, showing significant performance gains. Conversely, models with near-zero MD, such as *Qwen2.5-72B-Instruct* vs. *GPT-4o-mini* (MD = 0.088) and *Llama-3.1-8B-Instruct* vs. *GPT-4o-mini* (MD = 0.032), experience minimal or even negative performance changes after calibration. This finding highlights that if a model is already well-aligned with human preferences, additional calibration may have little effect or even introduce instability. To precisely quantify this relationship, we calculated the Pearson correlation between Mismatch Degree and its average performance improvement. The result is a high correlation of 0.747, confirming that MD is a practical tool for diagnosing mis-calibration in reward models.

## 6.2 Enhanced Robustness to Stylistic Variations

We observe that **CHARM** leads to consistent performance improvements on RM-Bench. This benchmark is particularly challenging because the differences between preferred and rejected responses are often subtle, and the responses exhibit diverse stylistic variations ranging from concise and straightforward to elaborate and well-formatted. Achieving higher scores on RM-Bench therefore suggests that the reward model becomes more robust to stylistic changes and better focuses on the factual reliability and coherence of responses rather than being misled by superficial patterns.

Based on this observation, we hypothesize that **CHARM** encourages the RM to prioritize semantic correctness and content reliability over superficial formatting that is frequently associated with certain policy models (Zhang et al., 2024b). For example, without calibration, an RM may incorrectly favor a response that is neatly formatted but factually incorrect, while undervaluing a simpler yet accurate response. After calibration with **CHARM**, however, the RM is better able to assign higher scores to the latter.

We conducted a analysis of the RM's sensitivity to stylistic variation. To remove potential confounding effects from score distribution, we first applied Z-score normalization across all RM-Bench samples. We then computed the average normalized scores that the RM assigned to the responses across three distinct style categories. The results show that after calibration, the variance of average Z-scores across chosen and rejected groups decreased from 0.123 to 0.099 and 0.058 to 0.049, reflecting **CHARM** implicitly enhances the robustness of reward models to stylistic variations.

## 6.3 From Explicit Patterns to Implicit Calibration

Based on our previous analysis, we conducted a further quantitative study on five specific stylistic patterns: emoji, length, bold, exclamation, and list. We calculated the frequency of these patterns appearing in chosen and rejected responses within the dataset we used. We introduce the Preference Ratio, defined as the ratio of a pattern's occurrences in chosen responses to its occurrences in rejected responses.

The analysis reveals that the most significant changes occurred in two strong stylistic features: emojis and bold formatting. Before calibration, the reward model exhibited a strong positive bias: its preference ratio for responses containing emojis was as high as 2.38, and for those with bold text, it was 1.25. After calibration with **CHARM**, this preference underwent a reversal. The calibrated model, in contrast, slightly discourages the use of emojis (ratio decreased to 0.62) and bold text (ratio decreased to 0.88).

For the other three stylistic patterns, the model's initial bias was not significant, with preference ratios of 1.05 (length), 1.17 (exclamation), and 0.84 (list). The calibration process had a smaller impact on the distribution of these patterns, with post-calibration ratios of 1.06, 1.21, and 0.99, respectively. It is worth noting that the calibration also corrected the model's slight negative bias against lists (from a ratio of 0.84) and brought it close to neutral (0.99).

This leads to a deeper insight: our proposed model-based calibration method can be understood mechanistically as a form of implicit calibration. The core point is that a reward model's preference for a specific policy model is, in essence, an inherited preference for patterns present in that model's outputs. However, these patterns extend far beyond the few explicit features we have listed; they more likely include a large number of subtle, implicit linguistic styles that are difficult to identify and enumerate through manual construction. By calibrating the overall score distribution through a global offset, we can systematically mitigate the reward model's over-reliance on these complex patterns without needing to explicitly identify and construct each biased pattern. This, in turn, achieves a more robust and fair evaluation.

## 7 Discussions

While our work focuses on discriminative reward models based on the Bradley-Terry model, we acknowledge the existence of alternative formulations, such as pairwise (Jiang et al., 2023) and generative (Zhang et al., 2024a) reward modeling. Our method focuses on constructing a debiased preference dataset rather than relying on the specific architecture of the reward model. To maintain

| Pattern | chosen | rejected | PR | chosen | rejected | PR |
|---------|--------|----------|-----|--------|----------|-----|
|         | *w/o calibration* | | | *w/ calibration* | | |
| Emoji | 357 | 150 | 2.38 | 193 | 313 | 0.61 |
| Length (avg.) | 1842 | 1752 | 1.05 | 1851 | 1742 | 1.06 |
| Bold | 16595 | 13313 | 1.24 | 13961 | 15941 | 0.87 |
| Excl. | 5541 | 4705 | 1.17 | 5609 | 4635 | 1.21 |
| List | 10546 | 12457 | 0.84 | 11438 | 11565 | 0.98 |

Table 4: Statistics of stylistic pattern occurrences in the calibration dataset. PR refers to the ratio of a pattern's occurrences in chosen responses to its occurrences in rejected responses.

simplicity and efficiency, we apply a single global offset to correct model preference bias. Since model preference can be interpreted as a mixture of category-specific distributions, it is natural to extend **CHARM** to a category-wise setting. We present experiments in the Appendix A.4 that explore this tradeoff, demonstrating that such an extension can further enhance **CHARM** performance.

To explore the pervasiveness of this bias, we conducted a preliminary probe into generative reward models (details in Appendix A.3). Our findings indicate that model preference bias is also present in GRMs, although models with stronger reasoning capabilities appear more robust. This suggests that the bias is not confined to scalar reward modeling; even generative RMs, often considered more interpretable, are susceptible to exploiting spurious implicit patterns. Reasoning models enhanced with reinforcement learning (Team, 2025; Guo et al., 2025) have demonstrated greater robustness and less influenced model preference bias, suggesting a promising direction for building more reliable reward models. However, GRM training paradigms are diverse and rapidly evolving (e.g., SFT, RL, prompting), making it nontrivial to determine how calibration should be integrated. While extending CHARM to GRMs is clearly valuable, adapting it effectively will require a careful examination of GRM-specific training pipelines. We leave this as a direction for future work.

This issue aligns with a broader challenge in the field: developing more reliable and interpretable reward models to prevent reward hacking. Our paper not only identifies a subtle yet significant bias but also introduces a straightforward and effective debiasing method. A key direction for future work is to further investigate the underlying mechanisms of model preference bias. Our additional experiments in Appendix A.1 suggest that this bias may be potentially associated with certain training paradigms, which we hope can provide insights for future research.

## 8 CONCLUSION

In this work, we identified a subtle yet consequential bias in widely used reward models, which we term model preference bias. This bias reflects a tendency for RMs to disproportionately overvalue responses from certain policy models, resulting in misaligned evaluations and unfair judgments. To address this issue, we introduced **CHARM**, a simple yet effective calibration method that leverages Elo scores from Chatbot Arena to construct debiased preference datasets.

Extensive experiments demonstrate that **CHARM** enhances the judging capability of reward models, yielding consistent performance gains on benchmarks such as RM-Bench and RewardBench, particularly in the chat domain. It also improves alignment with human preferences, mitigating model preference bias and generalizing effectively to unseen LLMs. Furthermore, we showed that **CHARM** increases robustness to stylistic variations, thereby reducing implicit pattern bias. This observation validates the effectiveness of our method and provides insights into the sources of model preference bias. Our findings highlight the need to further explore the mechanisms underlying model preference bias towards developing more interpretable, robust and reliable reward models.

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

# A APPENDIX

## A.1 MORE EXPERIMENTS ON MODEL PREFERENCE BIAS

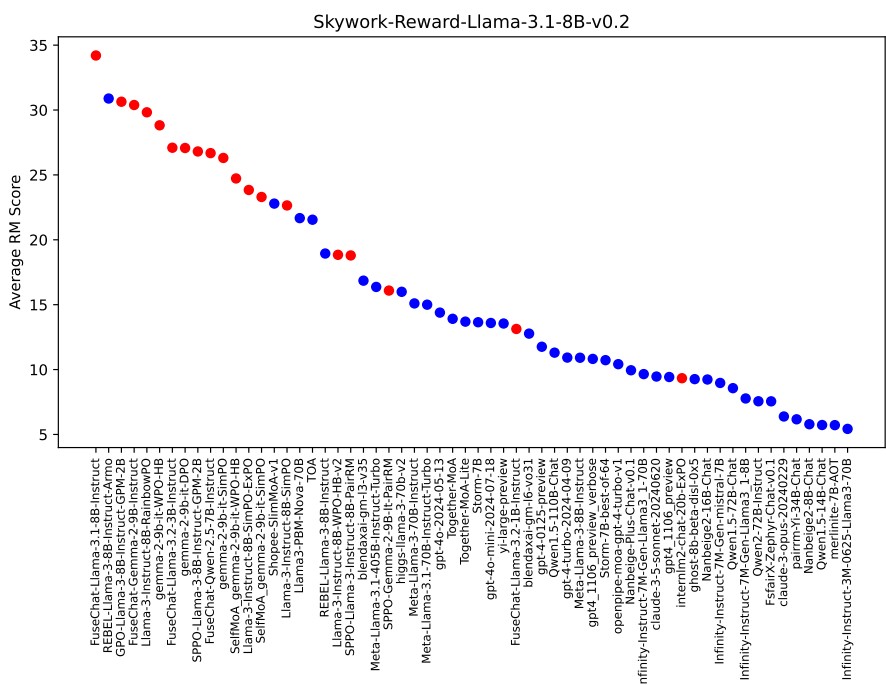

Figure 3: Score results of Skywork-RM on more policy models in the AlpacaEval dataset.

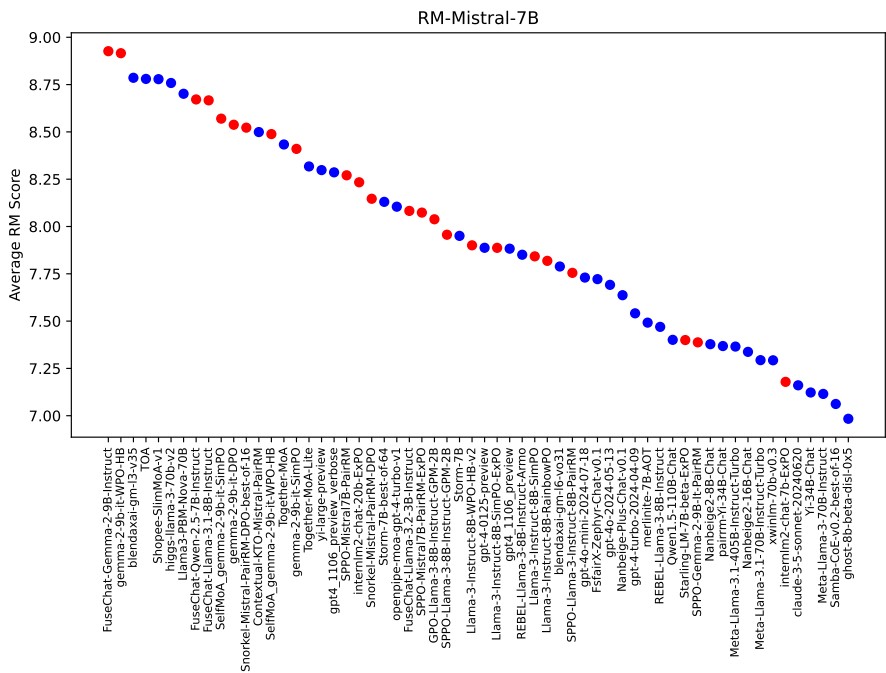

Figure 4: Score results of Mistral-RM on more policy models in the AlpacaEval dataset.

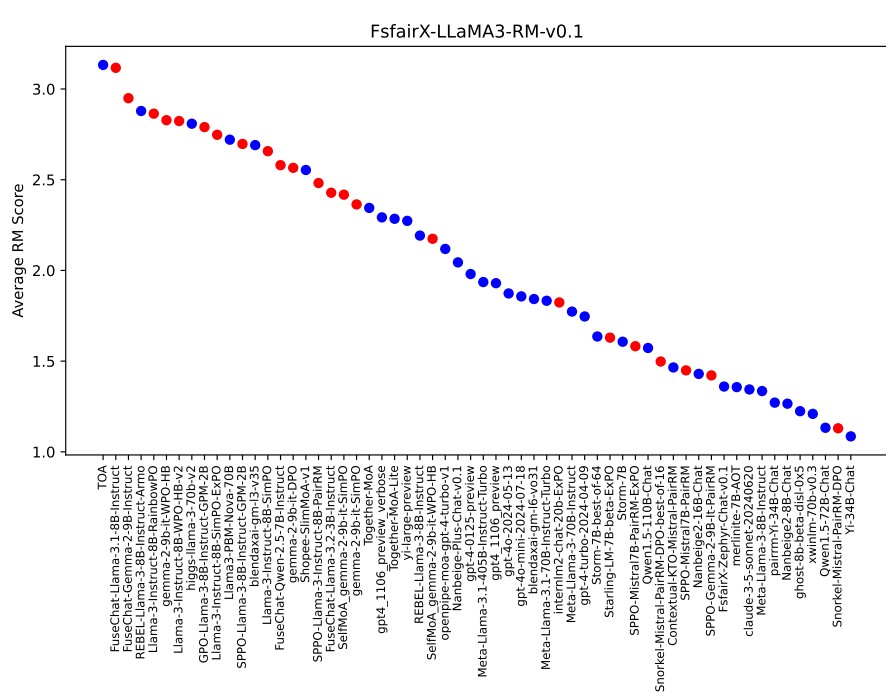

Figure 5: Score results of FsfairX-RM on more policy models in the AlpacaEval dataset.

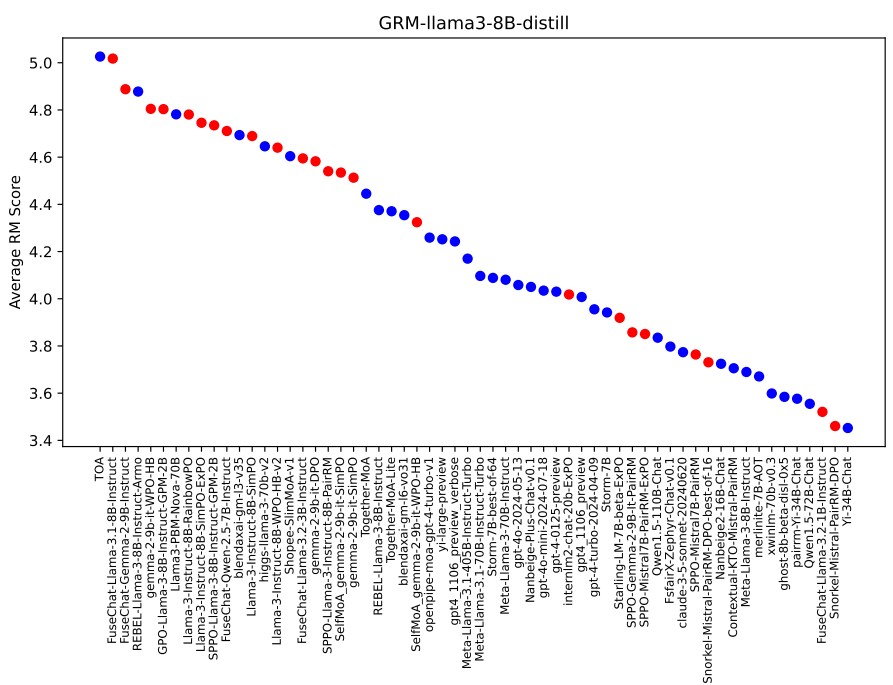

Figure 6: Score results of GRM-RM on more policy models in the AlpacaEval dataset.

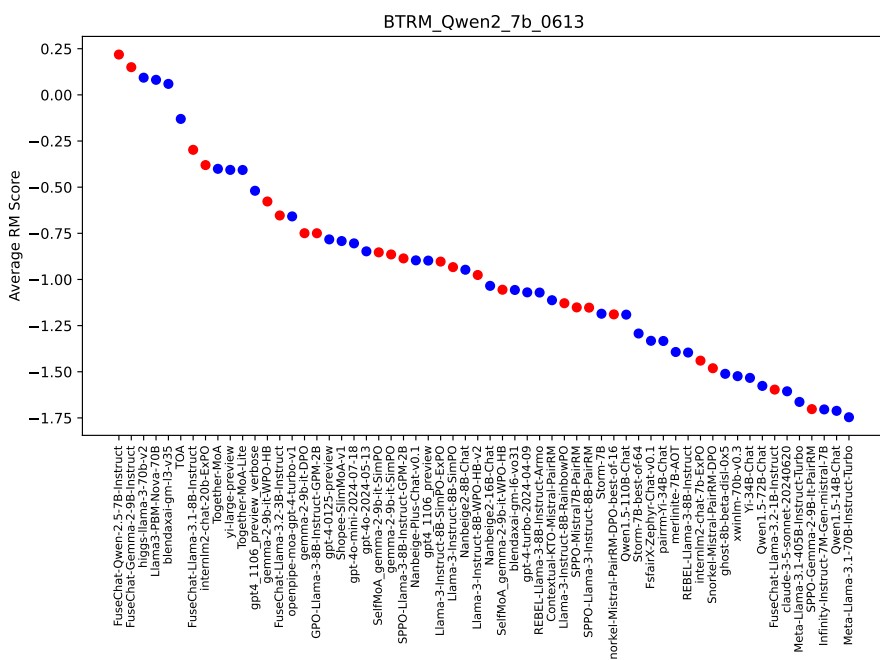

Figure 7: Score results of BTRM-RM on more policy models in the AlpacaEval dataset.

In Section 3, we compare the correlation between RM scores assigned to policy models and their Arena Elo scores. However, since many models listed on the AlpacaEval leaderboard have not participated in Chatbot Arena, their Elo scores are unavailable, preventing direct comparison with human preferences.

To further analyze model preference bias, we score responses from all 228 models available in the AlpacaEval dataset using different RMs. We then select the top 60 models ranked by Average RM scores. The results are illustrated in Figures 3–7, where models marked in red indicate those that have undergone preference optimization. These models are mostly around 7B parameters, significantly smaller than the top-ranking commercial models in Chatbot Arena. However, under RM evaluation, they exhibit a totally different ranking trend.

This finding suggests that the model preference bias exhibited by RMs is not a preference for specific individual model, but rather a systematic preference for a class of models. This, in turn, points to the formation mechanism of this bias being linked to common training methodologies or shared datasets utilized by this particular class of models. We hope this observation provides a valuable direction for future research into the origins of such biases.

A.2  POLICY MODEL DETAILS

In Section 3, we conducted an empirical study on a diverse set of policy models to probe model preference bias. This experiment relies on the Chatbot Arena Elo scores and the AlpacaEval win rates of the selected models. For completeness, we report the Elo scores and win rates used in our study in Table 5. Since the Chatbot Arena leaderboard is dynamically updated, these values may differ slightly from the latest results, but such discrepancies do not affect the overall conclusions of our experiments.

A.3  MODEL PREFERENCE BIAS IN GRMS

Generative Reward Models (GRMs) are a type of reward model that directly utilizes the judging capabilities of LLMs. Through techniques like Chain-of-Thought prompting (Wei et al., 2022), an LLM can be transformed into a vanilla GRM, also known as an "LLM-as-a-Judge". Furthermore, by fine-tuning on preference pairs, an LLM can be equipped with more robust reward modeling abilities

| Policy Models | Chatbot Arena Elo | AlpacaEval Winrate |
|---|---|---|
| gpt-4o-2024-05-13 | 1285 | 57.5 |
| gpt-4o-mini-2024-07-18 | 1272 | 50.7 |
| Meta-Llama-3.1-405B-Instruct-Turbo | 1269 | 39.3 |
| Qwen2-72B-Instruct | 1257 | 38.1 |
| gpt-4-turbo-2024-04-09 | 1256 | 55 |
| gpt4_1106_preview | 1250 | 50 |
| Meta-Llama-3.1-70B-Instruct-Turbo | 1248 | 38.1 |
| claude-3-opus-20240229 | 1247 | 40.5 |
| gemma-2-9b-it-SimPO | 1216 | 72.4 |
| claude-3-sonnet-20240229 | 1201 | 34.9 |
| gpt4_0314 | 1186 | 35.3 |
| Meta-Llama-3.1-8B-Instruct-Turbo | 1176 | 20.9 |
| gpt4_0613 | 1163 | 30.2 |
| mistral-large-2402 | 1157 | 32.7 |
| mistral-medium | 1148 | 28.26 |
| Mixtral-8x22B-Instruct-v0.1 | 1147 | 30.9 |
| gemini-pro | 1110 | 24.4 |
| OpenHermes-2.5-Mistral-7B | 1074 | 16.2 |
| Qwen1.5-7B-Chat | 1070 | 14.7 |
| gpt-3.5-turbo-1106 | 1068 | 19.3 |
| vicuna-13b | 1042 | 9.2 |
| gemma-7b-it | 1037 | 10.4 |
| vicuna-7b | 1005 | 6.3 |
| gemma-2b-it | 990 | 5.4 |

Table 5: Policy models used in our empirical study, along with their corresponding Chatbot Arena Elo scores and AlpacaEval win rates.

| GRMs | *deepseek-v3* **wins** | *gemma-2-9b-it-SimPO* **wins** | MD |
|---|---|---|---|
| Llama-3.1-8B | 500 | 499 | 0.296 |
| DeepSeek-R1-Distill-Qwen-7B | 570 | 365 | 0.143 |
| Skywork-Critic-Llama-3.1-8B | 338 | 662 | 0.525 |
| JudgeLRM-7B | 563 | 432 | 0.204 |
| Qwen3-8B | 762 | 233 | 0.07 |
| DeepSeek-R1 | 783 | 213 | 0.104 |

Table 6: Results of model preference bias on several GRMs.

| Category | Mismatch Degree |
|---|---|
| Code Generation | 0.736 |
| Creative Writing | 0.538 |
| Factual QA | 0.894 |
| Instruction Following | 0.411 |
| Math/Reasoning | 0.322 |
| Others | 0.875 |

Table 7: Mismatch Degree for each category.

(Zhu et al., 2023; Chen et al., 2025; Liu et al., 2025). GRMs are often considered more robust and interpretable than their scalar counterparts because the generative output format has a naturally higher information density at the cost of inference time.

This perceived robustness motivated our investigation into whether GRMs are also susceptible to the model preference bias identified in our work. To this end, we designed an experiment with three categories of models: (1) LLM-as-a-judge system utilizing CoT. (2) GRM fine-tuned on preference datasets. (3) Reasoning models enhanced with Reinforcement Learning.

For the experiment, we selected 1,000 prompts and let *DeepSeek-V3* and *Gemma-2-9B-it-SimPO* to generate responses. These response pairs were then judged by the different GRMs. To prevent positional bias, the order of the response pairs was shuffled before evaluation. The results are displayed in Table 6.

Both the LLM-as-a-judge system and the GRM fine-tuned on the preference dataset exhibited a discernible degree of model preference bias, with the *Skywork-Critic-Llama-3.1-8B* model showing the most significant bias. In contrast, the two reasoning models performed excellently, achieving near-perfect alignment with human preferences. This suggests that while even generative models are vulnerable to model preference bias, advanced training methods like RL that enhance a model's reasoning capabilities may be a promising direction for mitigating such biases.

### A.4 CATEGORY-WISE CHARM WITH MORE NUANCED CORRECTION

Model preference bias operates at the distributional level, and overall scoring behavior can be viewed as a mixture of category-specific distributions. While fine-grained, category-wise calibration is a natural extension, it introduces additional complexity in labeling and processing categories.

To explore this, we conducted an experiment using GPT-5 to categorize prompts in our calibration dataset into six types: **Math/Reasoning**, **Creative Writing**, **Code Generation**, **Factual QA**, **Instruction Following**, and **Others**. For each category, we computed the Mismatch Degree and optimized the offset between *Gemma-2-9b-it-SimPO* and *GPT-4o-mini* using Skywork-RM. **CHARM** was then applied separately within each category, and the calibrated datasets were merged for reward-model training. Results see Table 7 and Table 8.

Category-wise calibration yields an additional improvement of 0.9 points on RM-Bench, which is consistent with expectations. This indicates that for scenarios where maximizing performance is the primary goal, category-wise calibration can provide further gains, and **CHARM** naturally extends to this setting. However, as our main contribution emphasizes a simple and efficient calibration method, we adopt a single global offset in the main paper, highlighting the tradeoff between simplicity and fine-grained calibration.

### A.5 ROBUSTNESS TO CHATBOT ARENA ELO UPDATES

Chatbot Arena maintains a dynamic leaderboard where Elo ratings are continuously updated as new models enter and accumulate battle data. A concern is whether **CHARM** would require frequent

| Offset Type | RM-Bench | | | | |
|---|---|---|---|---|---|
| | Chat | Math | Code | Safety | *Avg* |
| Skywork-RM | 68.7 | 62.0 | 52.8 | 95.9 | 69.9 |
| *w/ global-wise CHARM* | 72.5 | 62.9 | 54.4 | 95.7 | 71.4 |
| *w/ category-wise CHARM* | 74.2 | 64.2 | 54.5 | 96.2 | 72.3 |

Table 8: RM-Bench performance of different offset type calibration.

| Calibration Version | Ref Elo | Over-valued Elo | RM-Bench | | | | |
|---|---|---|---|---|---|---|---|
| | | | Chat | Math | Code | Safety | *Avg* |
| Skywork-RM | | | 68.7 | 62.0 | 52.8 | 95.9 | 69.9 |
| CHARM (old Elo) | 1273 | 1216 | 73.9 | 62.4 | 53.9 | 95.8 | **71.5** |
| CHARM (newest Elo) | 1316 | 1278 | 72.5 | 62.9 | 54.4 | 95.7 | **71.4** |

Table 9: Robustness of CHARM to Elo rating updates. Despite substantial shifts in absolute Elo scores, calibration performance remains nearly identical.

recalibration as these ratings evolve. In this section, we evaluate the robustness of **CHARM** with respect to Elo rating updates.

In Equation 2, **CHARM** relies on Elo differences between model pairs rather than their absolute Elo values. When new models join the Arena, the absolute Elo scores of existing models may shift. However, the relative ranking among existing models tends to remain stable, especially between the strong–weak model pairs that **CHARM** targets.

To validate this robustness, we conducted an experiment using Elo scores from two different time periods. One retrieved during our initial experiments and another from the most recent leaderboard. Table 9 reports the RM-Bench performance comparison.

Despite substantial Elo shifts (+43 for the reference model and +62 for the over-valued model), the resulting performance difference is negligible. Both sets of Elo scores lead to nearly identical calibration improvements over the baseline. These results demonstrate that **CHARM** is robust to updates in the Elo leaderboard.

### A.6 COMPARISON WITH FINETUNING ON REAL ARENA BATTLES

A natural question is whether directly fine-tuning reward models on human preference data from Chatbot Arena could achieve similar bias mitigation effects as **CHARM**. In this section, we investigate this alternative approach and explain why CHARM is more effective.

Datasets like LMSYS-Chat-1M (Zheng et al., 2023a) or arena-human-preference-55k (Chiang et al., 2024) contain human preference battles across many models, but they lack sufficient samples involving specific over-valued models we need to correct, especially for smaller or research models (like Gemma-2-9b-it-SimPO we used). Instead **CHARM** uses Elo for they aggregate information across many battles and transitively through other models, providing more stable estimates than potentially sparse direct battles.

We conducted an experiment comparing **CHARM** against direct fine-tuning on Arena preference data. We finetuned Skywork-RM on arena-human-preference-55k, a dataset of real battles from Chatbot Arena for 1 epoch. Results see Table 10.

From the results we can see finetuning on arena-human-preference-55k leads to a substantial drop in RM-Bench performance and the Mismatch Degree remains high, indicating that general preference data does not address the specific model preference bias.

| Models | Mismatch Degree | RM-Bench | | | | |
|---|---|---|---|---|---|---|
| | | Chat | Math | Code | Safety | *Avg* |
| Skywork-RM | 0.639 | 68.7 | 62.0 | 52.8 | 95.9 | 69.9 |
| CHARM | 0.03 | 73.9 | 62.4 | 53.9 | 95.8 | 71.5 |
| Finetuning on arena-55k | 0.459 | 56.9 | 61.4 | 51.4 | 71.7 | 60.4 |

Table 10: Comparison of CHARM vs. finetuning on Arena battles. CHARM achieves both improved RM performance and reduced model preference bias, while Arena fine-tuning degrades performance and fails to mitigate bias.

| Method | RM-Bench | | | | |
|---|---|---|---|---|---|
| | Chat | Math | Code | Safety | *Avg* |
| Skywork-RM | 68.7 | 62.0 | 52.8 | 95.9 | 69.9 |
| Single model | 73.9 | 62.4 | 53.9 | 95.8 | **71.5** |
| Multiple models | 73.4 | 63.3 | 54.4 | 95.4 | **71.6** |

Table 11: Multiple models calibration results on RM-Bench.

## A.7 EXTEND CHARM TO MULTIPLE MODEL CALIBRATION

In practice, it may be necessary to calibrate more than one over-valued model at the same time. Thus, it is important to see whether **CHARM** can generalize to a multi-model setting. In this section, we extend **CHARM** to calibrate multiple over-valued models simultaneously.

We select four policy models, *Qwen2.5-72B-Instruct*, *Gemma-2-9b-it-SimPO*, *Llama-3.1-70B-Instruct*, and *GPT-4o-mini-2024-07-18*, and score their responses on Preference20K using Skywork-RM. Instead of performing pairwise calibration, we jointly optimize offsets for all models so that their calibrated scores match the entire Elo-derived win-rate matrix.

To be specific, we follow steps described below:

1. Construct the N × N target win-rate matrix $M_{\text{target}}$ from Elo scores, where

$$P(i \ vs. \ j) = \frac{1}{1 + 10^{(Elo_j - Elo_i)/400}}$$

2. Optimize global offsets $\{\Delta_1, \Delta_2, \ldots, \Delta_N\}$ by minimizing

$$\mathbf{MSE}(M_{current}, M_{target})$$

3. Construct the calibrated preference dataset using candidate responses from all four models, with calibrated scores $s'_i = s_i + \Delta_i$.

This multi-model extension evaluates whether **CHARM** can align an entire group of models with the human preference hierarchy derived from the Elo.

The result in Table 11 shows multiple model calibration achieves comparable performance to single-model calibration, demonstrating that joint optimization across multiple models maintains effectiveness.

## A.8 LLM USAGE

We used LLMs to polish the writing and assist in the implementation of parts of the codebase. All conceptual contributions, experiment designs, analyses were developed solely by the authors.

