# OpenReview forum: "CHARM: Calibrating Reward Models With Chatbot Arena Scores"
_ICLR.cc/2026/Conference — Submitted to ICLR 2026_

### Official Review · Reviewer_Wq5j · 2025-10-26

**Soundness:** 2
**Presentation:** 3
**Contribution:** 2
**Rating:** 4
**Confidence:** 3

**Summary:**

The submission compares the scores generated by popular reward models with the elo scores provided by chatbot arena and finds an inconsistency for certain models, which the submission interprets as misscalibration.
To prevent this issue, the authors to propose to first generate a dataset using the over-valued policies and reference policies and then use this dataset to construct a calibrated dataset of winning/losing reply pairs.
In experiment they show that finetuning an RM using this dataset can improve agreement with the reward bench dataset.

**Strengths:**

* The authors identify a new bias of popular reward models, particularly the preference for certain models beyond the probability implied by the chatbot arena elo. This is an interesting finding and is useful to be aware of when training RMs
 * The proposed method yields an improvement in RM accuracy on reward bench
 * The analysis of stylistic variations is interesting

**Weaknesses:**

* the entire paper is based on the premise that chatbot arena accurately reflects human preferences and that we want reward models to accurately correspond to chatbot arena scores, which is not sufficiently justified, particularly when actually using an RM for post-training
 * experiments also only consider correcting for the over-valuation of a single model (gemma-2-9b-it-SimPO). It is thus not clear whether  the results hold for other over-valued models, or in cases where we want to correct for multiple over-valued models at once
 * experiments show that calibration on chatbot arena leads to an improved reward accuracy on RM-bench. Experiments do not investigate whether using the calibrated RM is beneficial when preference-tuning a model. Recent research [2,3] has shown that reward accuracy on its own does not necessarily result in a better alignment of a post-trained model. Using the calibrated RM for post-training would significantly strengthen the experimental validation
 * terminology in the paper is inconsistent with the common usage of "calibration" in the context of supervised learning. Calibration usually refers to the error in probabilities $P_\theta(y|x)$ given by a classifier vs the true $P(y|x)$, in particular on a sample-wise level. The submission instead considers a class/model level bias of the reward model. Further the entire literature on classifier calibration is entirely ignored by the submission, see for example [1] for a survey


[1] Filho et al. "Classifier Calibration: A survey on how to assess and improve predicted class probabilities", Machine Learning 2023
[2] Razin et al. "What Makes a Reward Model a Good Teacher? An Optimization Perspective", NeurIPS 2025
[3] Chen et al. "The Accuracy Paradox in RLHF: When Better Reward Models Don't Yield Better Language Models", EMNLP 2024

**Questions:**

* Why was the target winrate P(O) determined from elo scores? Chatbot arena provides head-to-head winrates of model pairings, which directly yield P(O) per matchup instead of averaging over models. It seems like using this would be more direct.

---

> ### Author Response · Authors · 2025-11-16
> **Response to reviewer Wq5j (1/2)**
>
> > W1: the entire paper is based on the premise that chatbot arena accurately reflects human preferences and that we want reward models to accurately correspond to chatbot arena scores, which is not sufficiently justified, particularly when actually using an RM for post-training.
>
> Our premise is that Chatbot Arena is a good proxy for ranking models' relative performance because it aggregates general human preferences. We use this to identify and mitigate model preference bias in RMs.
>
> Our findings in Section 6 show that correcting model preference bias through CHARM leads to substantial style bias reduction. Style-biased RMs give unfair advantages based on formatting rather than genuine quality, which reflects the potential of applying RM for downstream applications.
>
> > W2: experiments also only consider correcting for the over-valuation of a single model (gemma-2-9b-it-SimPO). It is thus not clear whether the results hold for other over-valued models, or in cases where we want to correct for multiple over-valued models at once
>
> Thanks for your suggestions! In section 6.1, we replace the over-valued model to other model with varing mismatch degree, and we observe a correlation of performance gain and mismatch degree. We apologize for a labeling error in Table 2, the column header **Ref Models** should be **Over-valued Models** (the reference model was fixed as GPT-4o-mini). We think it can be a strong results to support the generalization of our method and can address your concern. We are conducting an additional experiment with some over-valued DPO-tuned models, and will report the results once the experiment is done. We also conducted experiments to see if we can fix over-valued models and use different reference models. We see consistent Improvements as all reference model choices lead to performance gains over the original RM, demonstrating robustness. Also there exhibits a correlation between MD and performance gains. This is consistent with our MD-as-indicator finding (Section 6.1) and proves stability and interpretability.
>
> **Table Re1**: RM-Bench performance of different choice of reference models.
> | **Over-valued Model** | **Reference Model**    | **MD** | **Chat** | **Math** | **Code** | **Safety** | **Avg** |
> |-----------------------|------------------------|--------|----------|----------|----------|------------|---------|
> | Original Skywork-RM   | -                      | -      | 68.7     | 62.0     | 52.8     | 95.9       | 69.9    |
> | gemma-2-9b-it-SimPO   | GPT-4o-mini            | 0.639  | 73.9     | 62.4     | 53.9     | 95.8       | 71.5    |
> | gemma-2-9b-it-SimPO   | Gemma-2-9b-it          | 0.582  | 70.7     | 63.2     | 54.6     | 94.7       | 70.8    |
> | gemma-2-9b-it-SimPO   | Gemma-2-27b-it         | 0.633  | 70.0     | 63.1     | 54.0     | 94.7       | 70.4    |
> | gemma-2-9b-it-SimPO   | Llama-3.1-8B-Instruct  | 0.506  | 71.4     | 64.5     | 53.7     | 95.9       | 71.4    |
> | gemma-2-9b-it-SimPO   | Llama-3.1-70B-Instruct | 0.675  | 72.2     | 64.7     | 55.0     | 95.2       | 71.8    |
> | gemma-2-9b-it-SimPO   | Qwen2.5-72B-Instruct   | 0.625  | 70.9     | 62.4     | 53.8     | 96.0       | 70.8    |
>
> We conduct an additional experiment to test whether CHARM can simultaneously calibrate multiple over-valued models. To be specific, we selected 4 policy models (Qwen2.5-72B-Instruct, gemma-2-9b-it-SimPO, Llama-3.1-70B-Instruct and  GPT-4o-mini-2024-07-18) then scored all responses on preference 20K with Skywork-RM.
>
> Instead of pairwise calibration, we jointly optimize offsets for all models to match the full Elo-derived winrate matrix:
>
> 1. Construct N×N target winrate matrix from Elo scores where $\mathbb{P}(i > j) = \frac{1}{1+10^{((Elo_j - Elo_i) / 400)}}$
> 2. Find offsets $\Delta_1, \Delta_2, ..., \Delta_n$ by minimizing MSE(Current_Winrate_Matrix, Target_Winrate_Matrix)
> 3. Construct calibrated preference dataset using candidate responses from all four models, where calibrated scores are $s^\prime_i = s_i + \Delta_i$
>
> The result shows it achieves comparable performance to single-model calibration, demonstrating that joint optimization across multiple models maintains effectiveness.
>
> **Table Re2**: RM-Bench performance of calibrating multiple models at once.
> | **Calibration Method** | **Models** | **Chat** | **Math** | **Code** | **Safety** | **Avg** |
> |------------------------|------------|----------|----------|----------|------------|---------|
> | Original Skywork-RM    | -          | 68.7     | 62.0     | 52.8     | 95.9       | 69.9    |
> | Single model           | 1          | 73.9     | 62.4     | 53.9     | 95.8       | 71.5    |
> | Multiple models        | 4          | 73.4     | 63.3     | 54.4     | 95.4       | 71.6    |

---

> ### Author Response · Authors · 2025-11-16
> **Response to reviewer Wq5j (2/2)**
>
> > W3: experiments show that calibration on chatbot arena leads to an improved reward accuracy on RM-bench. Experiments do not investigate whether using the calibrated RM is beneficial when preference-tuning a model. Recent research [2,3] has shown that reward accuracy on its own does not necessarily result in a better alignment of a post-trained model. Using the calibrated RM for post-training would significantly strengthen the experimental validation.
>
> We appreciate this important concern and acknowledge it as a limitation of our current work. The consistency between reward model benchmark performance and downstream RLHF performance has been questioned for a long time. In RM-Bench, the authors demonstrate a correlation between their benchmarks and downstream task outcomes.
>
> We agree that, as a component within the RLHF pipeline, a reward model should ideally be evaluated in the context of post-training. However, how to rigorously measure this alignment is actually notoriously difficult. Given this, we choose to report reward model benchmark performance, as it provides a relatively fair and standardized way to demonstrate the effectiveness of our method.
>
> > W4: terminology in the paper is inconsistent with the common usage of "calibration" in the context of supervised learning. Calibration usually refers to the error in probabilities given by a classifier vs the true , in particular on a sample-wise level. The submission instead considers a class/model level bias of the reward model. Further the entire literature on classifier calibration is entirely ignored by the submission, see for example [1] for a survey
>
> Thank you for raising this issue. Our usage of the term *calibration* follows prior work on reward model calibration [1], where *calibration* refers to debiasing a reward model so that its outputs better align with human preferences. This notion of calibration operates at the model or class level, consistent with the scope of reward modeling.
>
> While this differs from the conventional usage of *calibration* in supervised learning, we note that the two settings share a conceptual similarity: both aim to align a model’s outputs with a corresponding ground truth (in classifier calibration, true class probabilities; in reward model calibration, the “golden” preference).
>
> We acknowledge that our current draft does not sufficiently discuss the extensive literature on classifier calibration. We will add a subsection in the related work to clarify the distinction and connection between these two notions of calibration, and to provide appropriate citations to the classifier calibration literature.
>
> [1] Huang Z, Qiu Z, Wang Z, et al. "Post-hoc Reward Calibration: A Case Study on Length Bias", ICLR 2025
>
> > Q1: Why was the target winrate P(O) determined from elo scores? Chatbot arena provides head-to-head winrates of model pairings, which directly yield P(O) per matchup instead of averaging over models. It seems like using this would be more direct.
>
> Direct head-to-head battle data is not available for many model pairs, especially for smaller or research models (like Gemma-2-9b-it-SimPO we used). Even when direct battle data exists, it often suffers from sparse sampling. Instead we use Elo for they aggregate information across many battles and transitively through other models, providing more stable estimates than potentially sparse direct battles. For example, even without direct A vs. B battles, Elo can estimate their relative strength through shared opponents (A beats C, C beats B → A vs. B).
>
> ---
>
> Thanks again for your careful review and insightful suggestions. We hope our responses adequately address your concerns and help clarify and strengthen our contributions. If you have any further questions or would like us to elaborate on any point, please feel free to let us know. We will do our best to respond.

---

> > ### Comment · Reviewer_Wq5j · 2025-11-20
> >
> > Thank you for the rebuttal.
> >
> > The additional experiments with multi-model calibration are promising and I appreciate the explanation for why head-to-head winrates could not be used.
> >
> > While I understand that actually post-training a model with RL is more compute intense than training a reward model, I do believe it would significantly strengthen the contribution of this paper.
> >
> > Further, if the current method mainly removes certain stylistic biases of the reward model, this is nice but there are alternative ways to achieve this, for example by adding a bit of additional preference training data that includes the style of interest.
> >
> > As such, I do not currently see evidence that this method will be useful in practice, and thus will keep my score.

---

> > > ### Author Response · Authors · 2025-11-22
> > > **Thanks for your response! More results on post-training with RM.**
> > >
> > > Thanks for your valuable feedback! We fully recognize the importance of evaluating reward models in a post-training setting. To address this, we conducted an additional experiment in which we trained a DPO model using a calibrated reward model. Since model preference bias may originate from biased preference-learning datasets in our analysis, offline DPO provides an ideal setting for testing the effectiveness of CHARM.
> > >
> > > We used GPT-4o-mini and Gemma-2-9b-it-SimPO responses on Preference20K and obtained reward scores from both the original Skywork-RM and the CHARM-calibrated Skywork-RM. We then constructed two DPO datasets from these preference pairs and trained **Qwen2.5-7B-Instruct** separately on each. We evaluated the resulting models on IFEval, and the results are shown in Table Re3. Models trained with CHARM-calibrated RM consistently outperform those trained with the original RM and baseline model across all IFEval metrics.
> > >
> > > **Table Re3**: IFEval results of DPO variants.
> > > | **Model**                | **Inst Loose** | **Inst Strict** | **Prompt Loose** | **Prompt Strict** |
> > > |----------------------------|----------------|-----------------|------------------|-------------------|
> > > | Qwen2.5-7b-it              | 72.06          | 67.51           | 62.11            | 56.75             |
> > > | DPO on Skywork-RM          | 72.30          | 67.87           | 62.29            | 56.19             |
> > > | DPO on CHARM-calibrated RM | 74.46          | 68.47           | 64.88            | 57.67             |
> > > | Improvement                | +2.16          | +0.6            | +2.59            | +1.48             |
> > >
> > > As for your second concern, we would like to clarify our core claim that model preference bias is an inherited preference for patterns present in certain models’ outputs, and these patterns extend far beyond the explicit stylistic features we listed. The explicit stylistic bias we listed serve only as a case study because they are easy to observe and quantify. However, as we emphasize in Section 6.3, the actual biased patterns are likely far more subtle, high-dimensional, and difficult to enumerate.
> > >
> > > By calibrating the overall score distribution, CHARM mitigates the reward model’s over-reliance on these complex, implicit patterns without requiring us to explicitly identify or manually construct each biased feature. We believe this ability to correct distribution-level bias in a model-agnostic manner is a key contribution of our method, which is more efficient than adding more preference training data related to certain bias.
> > >
> > > Thanks again for your valuable advice. Look forward to further discussions!

---

### Official Review · Reviewer_8Hwd · 2025-11-01

**Soundness:** 3
**Presentation:** 3
**Contribution:** 2
**Rating:** 4
**Confidence:** 4

**Summary:**

This paper identifies model preference bias—a systematic tendency of scalar reward models (RMs) to overrate responses from certain policy models (e.g., Gemma-2-9b-it-SimPO), even when those models perform modestly on human-aligned benchmarks like Chatbot Arena. To address this, the authors propose CHARM, a calibration method that leverages publicly available Elo scores from Chatbot Arena to adjust RM scoring via a learned offset Δ, thereby constructing debiased preference pairs for fine-tuning. Experiments show that CHARM improves RM performance on RM-Bench and RewardBench (especially in the Chat domain), better aligns RM judgments with human preferences, enhances robustness to stylistic variations, and generalizes to unseen models.

**Strengths:**

1. Studies an important and underexplored problem: calibrating RMs using real-world human preference signals (Chatbot Arena Elo).
2. Demonstrates clear empirical gains for small-to-medium scalar RMs after calibration, with thorough analysis of bias reduction and generalization.

**Weaknesses:**

1. The method fits RM scores to Elo-derived win rates via MSE. But why not train directly on Chatbot Arena’s raw battle data or use Elo-predicted probabilities as soft labels? That would seem more direct and principled.
2. Results are limited to scalar RMs. Given the growing shift toward Generative Reward Models (GRMs), it’s crucial to show whether CHARM can also improve GRMs (e.g., by relabeling preference data used to train LLM-as-a-judge systems).
3. The Elo scores used—are they the standard raw Arena Elo or style-controlled (e.g., length-normalized)? If the latter, the observed debiasing might stem from style control rather than the calibration mechanism itself. Clarification is needed.

**Questions:**

see weakness

---

> ### Author Response · Authors · 2025-11-16
> **Response to reviewer 8Hwd (1/2)**
>
> > W1: The method fits RM scores to Elo-derived win rates via MSE. But why not train directly on Chatbot Arena’s raw battle data or use Elo-predicted probabilities as soft labels? That would seem more direct and principled.
>
> Thanks for your questions. We didn’t train directly on Arena’s raw battles because the publicly available Arena datasets lack sufficient battles between many model pairs, especially those involving over-valued models. For example, our used model pair (Gemma-2-9b-it-SimPO vs. GPT-4o-mini) has few battles in the available Arena datasets. By using Elo scores, we can compute expected win rates for any model pair with Elo ratings, even if they haven't directly battled. This makes our method more scalable and flexible, not limited to model pairs with existing battle data. Also, training on raw battle datasets is more like pretraining a reward model to learn general human preferences from scratch, which is certainly valuable but falls outside the scope of our work as we aim to mitigate model preference bias.
>
> We also conduct an additional experiment of finetuning Skywork-RM on arena-human-preference-55k for one epoch. We evaluated the finetuned model on RM-Bench and computed the Mismatch Degree to quantify its model preference bias. Results see Table Re1. The results show that finetuning on arena-human-preference-55k not only leads to a substantial drop in reward-model performance but also leaves the mismatch degree high, indicating that model preference bias is not effectively mitigated. In contrast, CHARM yields both improved RM-Bench performance and a significant reduction in mismatch degree.
>
> As for elo-predicted probabilities as soft labels, we uses the standard Bradley-Terry loss with hard preference labels, which needs strict preference pairs $(y_+, y_-)$. We don’t want to change existing RM training pipelines, so we prefer a more standard way to perform the calibration.
>
> **Table Re1**: RM-Bench performance of training on direct arena battles data.
> | **Calibration Version**             | **Mismatch Degree** | **Chat** | **Math** | **Code** | **Safety** | **Avg** |
> |-------------------------------------|:-------------------:|:--------:|:--------:|:--------:|:----------:|:-------:|
> | Original Skywork-RM                 | 0.639               | 68.7     | 62.0     | 52.8     | 95.9       | 69.9    |
> | CHARM                               | 0.03                | 73.9     | 62.4     | 53.9     | 95.8       | 71.5    |
> | train on arena-human-preference-55k | 0.459               | 56.9     | 61.4     | 51.4     | 71.7       | 60.4    |
>
> > W2: Results are limited to scalar RMs. Given the growing shift toward Generative Reward Models (GRMs), it’s crucial to show whether CHARM can also improve GRMs (e.g., by relabeling preference data used to train LLM-as-a-judge systems).
>
> We appreciate this important concern about the growing trend toward GRMs. We have conducted preliminary investigations into model preference bias in GRMs, which we have reported in **Appendix A.3.** We found Model preference bias still persists in GRMs, especially for weaker models like Skywork-GRM and judgeLRM. Stronger reasoning models show greater robustness with a relatively low MD.
>
> However, GRM training paradigms are diverse and evolving (SFT, RL, prompt engineering, etc.), which makes it unclear how to best integrate calibration. Extending CHARM to GRMs is valuable. Our Appendix A.3 establishes the motivation (bias exists). Adapting CHARM requires dedicated investigation of GRM-specific training pipelines and output formats, which we leave for future work. We will expand this discussion in the revision manuscript.

---

> ### Author Response · Authors · 2025-11-16
> **Response to reviewer 8Hwd (2/2)**
>
> > W3: The Elo scores used—are they the standard raw Arena Elo or style-controlled (e.g., length-normalized)? If the latter, the observed debiasing might stem from style control rather than the calibration mechanism itself. Clarification is needed.
>
> CHARM computes expected win rates from Elo differences (Equation 2) and uses them to mitigate model preference bias. Even if style control shifts individual Elo scores, the relative gap remains largely stable. Take the model pair we use (Gemma-2-9b-it-SimPO vs. GPT-4o-mini) as an example, Style control changes the gap by 21 points, translating to only 3.1% difference in expected win rate.
>
> **Table Re2**: Comparison of raw and style-controlled elo.
> | **Elo Type**     | **Gemma-2-9b-it-SimPO** | **GPT-4o-mini** | **Gap** | **Expected Win Rate** |
> |------------------|-----------|------------|---------|-----------------------|
> | Raw              | 1278      | 1316       | 38      | 44.5%                 |
> | Style-controlled | 1228      | 1287       | 59      | 41.5%                 |
>
> In practice, we use style-controlled Elo scores as this is the default of Chatbot Arena, but the observed debiasing does not stem from style control. We conduct an extra experiment of calibrating Skywork-RM using both Elo types. Table Re3 show that using raw elo scores also leads to performance gains, indicating the effectiveness of debiasing.
>
> **Table Re3**: RM-Bench performance of using raw and style-controlled elo.
> | **Elo Type**        | **Chat** | **Math** | **Code** | **Safety** | **Avg** |
> |---------------------|----------|----------|----------|------------|---------|
> | Original Skywork-RM | 68.7     | 62.0     | 52.8     | 95.9       | 69.9    |
> | Style-controlled    | 72.5     | 62.9     | 54.4     | 95.7       | 71.4    |
> | Raw                 | 73.7     | 63.0     | 54.9     | 95.7       | 71.8    |
>
> ---
>
> Thanks again for your careful review and insightful suggestions. We hope our responses adequately address your concerns and help clarify and strengthen our contributions. If you have any further questions or would like us to elaborate on any point, please feel free to let us know. We will do our best to respond.

---

> ### Comment · Reviewer_8Hwd · 2025-11-26
>
> Thank you for the detailed response. While it clarifies some points, my main concerns regarding GRM remain. Therefore, I will keep my score.

---

> > ### Author Response · Authors · 2025-11-26
> >
> > Thanks for your reply. We want to make a few points:
> >
> > 1. In current RLHF pipelines, the most widely used reward models are still scalar RMs, while GRMs most act as LLM-as-a-judge, relying heavily on the model’s internal reasoning and evaluation capabilities. As we showed in the Appendix, LLMs that acquire strong reasoning ability through RL tend to exhibit smaller model preference bias, we argue that for GRMs, improving the model’s reasoning ability is a more effective way to mitigate bias, unlike scalar RMs, whose decision boundaries can be updated through continual learning on a small amount of preference pairs.
> > 2. Scalar RMs and GRMs follow different training paradigms and mechanisms, and therefore often require different calibration strategies. “Relabeling preference data used to train LLM-as-a-judge systems” that you mentioned is not applicable to CHARM, because CHARM requires the reward model to first produce scalar scores for responses so that an offset can be applied. In contrast, LLM-as-a-judge GRMs perform pairwise comparisons, producing only discrete preference outcomes, which makes it impossible to compute the offset needed for CHARM.
> > 3. Since RM and GRM use different fundemental training algorithms, so we cannot agree that it is a weakness that our method for RM cannot be applicable to GRM. This inapplicability applies to most RM works. The fact that CHARM is not directly applicable to GRMs **does not undermine its contribution**.
> >
> > Look forward to further discussions!

---

### Official Review · Reviewer_B6Db · 2025-11-01

**Soundness:** 3
**Presentation:** 2
**Contribution:** 3
**Rating:** 4
**Confidence:** 4

**Summary:**

This paper proposes CHARM, a method to alleviate reward hacking via calibrating the Reward Models (RM) with the Elo rating (Human Preference). Specifically, they utilize the Elo Rating from the public Chatbot Arena and calibrate the reward model's training dataset to align the win-rate estimated from the reward scores with the expected win-rate from the Elo rating. They then fine-tune the reward model to calibrate the reward score to mitigate the Model Preference Bias. Experiments show that the calibrated RM is more aligned with human preference/Elo rating and obtains performance gains against existing RM benchmarks.

**Strengths:**

1. The motivation is clear and interesting. Using the Elo rating to calibrate the reward models to make the reward model more aligned with human preferences is sound.
2. The formulation of the method part is clear and easy to follow. The authors first analyze the correlation between Elo rating and reward model scores, identifying the potential Model Preference Bias, and then explain how they address it (via calibration).
3. Experiments show that using the calibrated dataset to fine-tune the reward model can improve its performance.

**Weaknesses:**

1. My main concern is that CHARM is not scalable. Each reward models need their own calibrated dataset for further fine-tuning. Besides, as the elo rating is updated frequently, do you need to recalibrate when the Chatbot Arena is updated? (I may misunderstand the method; point me out if I'm wrong.)
2. While over-valued and reference models play an important role in calibration, how to select them is not adequately elaborated in the paper. This hinders the critical understanding of CHARM, i.e., how to select or identify the potentially over-valued models in a more systematic manner.

**Questions:**

1. Instead of using the calibrated dataset, what if we directly use the preference data collected from the chatbot arena (e.g., LMSYS-Chat-1M) to fine-tune the reward model?
2. How to select the over-valued and reference models?
3. Can you explain the difference between the calibrated data curated by CHARM and human-annotated preference data?
4. Have you considered using training-free calibration methods, such as temperature scaling, for calibrating the scalar RM?
5. Does the update of the chatbot arena's Elo rating affect the calibration performance?

---

> ### Author Response · Authors · 2025-11-16
> **Response to Reviewer B6Db (1/2)**
>
> > W1.1: My main concern is that CHARM is not scalable. Each reward models need their own calibrated dataset for further fine-tuning.
>
> Thanks for your question. We assume you are concerned about the efficiency of our method. There may be a misunderstanding about our method's workflow. CHARM does not require generating separate datasets for each RM. We generate a shared response pool using different policy models on sampled instructions. The response generation step is done once and shared across all RMs. It serves as the foundation for calibrating all RMs. Each RM only needs to: (1) score the pre-generated responses and (2) apply CHARM to construct the calibrated preference dataset. This process is very lightweight and efficient. Given the use of a shared response pool, we may say our method is actually quite scalable instead.
>
> > W1.2: Besides, as the elo rating is updated frequently, do you need to recalibrate when the Chatbot Arena is updated?
> > Q5: Does the update of the chatbot arena's Elo rating affect the calibration performance?
>
> Arena Elo ratings are updated as new models join. However, we argue that frequent recalibration is unnecessary. CHARM computes expected win rates from Elo differences (Equation 2) and uses them to mitigate model preference bias. While *absolute Elo* values shift when new models enter, the *relative ranking* of existing models remains stable.
>
> To directly test your concern, we recalibrated Skywork-RM using the most recent Arena Elo scores and compared performance to our original calibration. Results see Table Re1.
>
> Despite substantial Elo shifts (+43 for reference, +62 for over-valued model), the performance difference is negligible. Both calibrations provide nearly identical improvements over the baseline. This demonstrates that CHARM is robust to Elo updates, and frequent recalibration is unnecessary. We nonetheless thank you for the question, and we can include this in the appendix.
>
> **Table Re1**: RM-Bench performance of different Elo version used.
> | **Elo Version** | **Ref Elo** | **Overvalued Elo** | **Chat** | **Math** | **Code** | **Safety** | **Avg** |
> |-------------------------|:-----------:|:------------------:|:--------:|:--------:|:--------:|:----------:|:-------:|
> | Original Skywork-RM     | -           | -                  | 68.7     | 62.0     | 52.8     | 95.9       | 69.9    |
> | CHARM (old Elo)         | 1273        | 1216               | 73.9     | 62.4     | 53.9     | 95.8       | 71.5    |
> | CHARM (newest Elo)      | 1316        | 1278               | 72.5     | 62.9     | 54.4     | 95.7       | 71.4    |
>
> > W2: While over-valued and reference models play an important role in calibration, how to select them is not adequately elaborated in the paper. This hinders the critical understanding of CHARM, i.e., how to select or identify the potentially over-valued models in a more systematic manner.
> > Q2: How to select the over-valued and reference models?
>
> In our practice, our selection pipeline is as follow:
>
> 1. Select ~25 policy models with varying capabilities (reflected by Elo scores) and collect their responses on sampled instructions from AlpacaEval.
> 2. Apply the reward model being evaluated to score all responses, producing a scoring profile for each policy model (as shown in Figure 1).
> 3. Select a strong reference model (we choose GPT-4o-mini) and compute the Mismatch Degree for each policy model against this reference.
> 4. Select the model pair with highest MD for calibration (we choose Gemma-2-9b-it-SimPO with MD=0.639 in our case).
>
> To make this process more systematic and accessible, we will open-source the selection pipeline and expand the model pool. This will enable the community to systematically detect and mitigate model preference bias in any reward model. We will also add more clear description of the selection pipeline in the revised manuscript.

---

> ### Author Response · Authors · 2025-11-16
> **Response to reviewer B6Db (2/2)**
>
> > Q1: Instead of using the calibrated dataset, what if we directly use the preference data collected from the chatbot arena (e.g., LMSYS-Chat-1M) to fine-tune the reward model?
>
> Thanks for your insightful questions. There are several reasons why we use calibrated datasets rather than direct Arena data.
>
> Datasets like LMSYS-Chat-1M or arena-human-preference-55k contain human preference battles across many models, but they lack sufficient samples involving the specific over-valued models we need to correct. By generating responses from any model pair on-demand (no human battles needed), we can make CHARM more scalable and practical.
>
> CHARM is designed to mitigate model preference bias in existing RMs. Training on datasets like LMSYS-Chat-1M is more like pretraining a reward model to learn general human preferences from scratch, which is certainly valuable but falls outside the scope of our work.
>
> We also conduct an additional experiment of finetuning Skywork-RM on arena-human-preference-55k for one epoch. We evaluated the finetuned model on RM-Bench and computed the Mismatch Degree to quantify its model preference bias. The results show that finetuning on arena-human-preference-55k not only leads to a substantial drop in reward-model performance but also leaves the mismatch degree high, indicating that model preference bias is not effectively mitigated. In contrast, CHARM yields both improved RM-Bench performance and a significant reduction in mismatch degree.
>
> **Table Re2**: RM-Bench performance of training on direct arena battles data.
> | **Calibration Version**             | **Mismatch Degree** | **Chat** | **Math** | **Code** | **Safety** | **Avg** |
> |-------------------------------------|:-------------------:|:--------:|:--------:|:--------:|:----------:|:-------:|
> | Original Skywork-RM                 | 0.639               | 68.7     | 62.0     | 52.8     | 95.9       | 69.9    |
> | CHARM                               | 0.03                | 73.9     | 62.4     | 53.9     | 95.8       | 71.5    |
> | train on arena-human-preference-55k | 0.459               | 56.9     | 61.4     | 51.4     | 71.7       | 60.4    |
>
> > Q3: Can you explain the difference between the calibrated data curated by CHARM and human-annotated preference data?
>
> Human-annotated preference dataset leverages direct human judgments where annotators compare two responses and select the better one. This reflects the specific annotators' preferences and criteria at the time of labeling.
>
> CHARM is designed to mitigate model preference bias. Given our premise that Arena Elo provides a relatively accurate proxy for human preferences, CHARM produces preference labels that approximate what human annotators would assign if they were to label the same model-pair responses at scale. Specifically, if human annotators were to label a preference dataset with responses from two policy models, the resulting win rate should approximate the Elo-derived win rate. CHARM achieves this alignment through calibration, but with a more efficient way.
>
> Some other human-annotated preference dataset may focus on some other specific criteria, for example, helpness or safety. Those preference datasets are designed to instill particular values or behaviors in RMs, which is different from the objective of our method.
>
> > Q4: Have you considered using training-free calibration methods, such as temperature scaling, for calibrating the scalar RM?
>
> Thank you for this suggestion. To our knowledge, temperature scaling also requires a calibration dataset to optimize the temperature parameter T. Do you mean we could apply temperature scaling instead of fine-tuning the RM?
>
> If so, we believe this is an alternative training method, but it would not undermine CHARM's core contribution. We would happy to add this experiment if we understood correctly.
>
> ---
>
> Thanks again for your careful review and insightful suggestions. We hope our responses adequately address your concerns and help clarify and strengthen our contributions. If you have any further questions or would like us to elaborate on any point, please feel free to let us know. We will do our best to respond.

---

> ### Comment · Reviewer_B6Db · 2025-11-27
>
> Thanks for your response. It helps me better understand how CHARM works in practice.
>
> It is also suggested to add these discussions after the rebuttal to improve the clarity and completeness of your manuscript.

---

> > ### Author Response · Authors · 2025-11-28
> > **Thanks for your reply and updating score!**
> >
> > Thanks for your reply! We appreciate the time and effort you have dedicated to reviewing our work. We promise to include these discussions in the revised version. If you have any further questions or concerns, we remain dedicated to addressing them with the utmost eagerness.

---

### Official Review · Reviewer_q9aZ · 2025-11-08

**Soundness:** 2
**Presentation:** 3
**Contribution:** 2
**Rating:** 6
**Confidence:** 3

**Summary:**

This paper identifies "Model Preference Bias" in reward models, which is a systematic tendency to assign disproportionately high scores to responses from certain policy models. To mitigate this, the authors propose CHARM, a calibration method that uses Elo ratings from Chatbot Arena as a proxy for ground-truth human preferences. CHARM computes a global score offset Δ for an "over-valued" model to align the RM's empirical win rate with the Elo-derived expected win rate. The calibrated RMs show improved performance on RM-Bench and RewardBench, better alignment with Chatbot Arena rankings, and enhanced robustness to stylistic variations. The authors also introduce a "Mismatch Degree" metric to quantify bias and demonstrate generalization to unseen models.

**Strengths:**

- The paper articulates a subtle but important bias (i.e., "Model Preference Bias") that has been overlooked in reward modeling literature.

- CHARM is straightforward to implement, requiring only a single additive offset per model. This simplicity makes it attractive for practitioners who cannot afford complex retraining pipelines. The method leverages readily available Chatbot Arena data, which is a clever use of existing resources.

- The consistent improvements across multiple RMs (Table 1) and the reduction in Mismatch Degree (Figure 2) are compelling. The analysis of stylistic patterns (Table 3) provides plausible evidence that CHARM mitigates implicit biases beyond just model-specific ones.

**Weaknesses:**

- Oversimplified Calibration Mechanism: The core assumption that a *single global additive offset* Δ can correct complex, instruction-dependent biases is theoretically questionable. Model preference bias likely manifests differently across prompt categories (e.g., coding vs. creative writing), yet CHARM applies a uniform correction. The paper provides no analysis of whether Δ varies by domain or instruction type, nor ablations showing why a more nuanced correction (e.g., instruction-specific offsets) is unnecessary.

- Weak Ground Truth Assumption: The method treats Chatbot Arena Elo scores as absolute ground truth for human preferences, but these scores have well-documented limitations: they reflect a specific user population, are influenced by positional bias, and conflate multiple criteria (helpfulness, safety, style). The paper doesn't address how Chatbot Arena's biases might propagate into CHARM. For example, if Arena users prefer verbose responses, CHARM might inadvertently bake this length bias *into* the RM rather than remove it.

- Limited Calibration Scope: The main experiments calibrate using only *one* over-valued model (Gemma-2-9b-it-SimPO) and *one* reference model (GPT-4o-mini). This is a major limitation: (1) The paper claims bias is systematic across "preference-optimized" models, but only demonstrates calibration on a single instance. Testing with multiple over-valued models (e.g., DPO-tuned, PPO-tuned) is essential to validate generalizability. (2) The offset Δ is fundamentally tied to the choice of π_R. The paper fails to explore how Δ changes with different references (e.g., a weaker model like Llama-3-3.1-8B vs. GPT-4o). This raises questions about the stability and interpretability of the calibration.

**Questions:**

What is the reward model in line 240 to produce the uncalibrated preference dataset? Will the choice of this reward model influence the final results?

---

> ### Author Response · Authors · 2025-11-16
> **Response to Reviewer q9aZ (1/2)**
>
> > W1: The core assumption that a single global additive offset Δ can correct complex, instruction-dependent biases is theoretically questionable. Model preference bias likely manifests differently across prompt categories (e.g., coding vs. creative writing), yet CHARM applies a uniform correction. The paper provides no analysis of whether Δ varies by domain or instruction type, nor ablations showing why a more nuanced correction (e.g., instruction-specific offsets) is unnecessary.
>
> Thanks for your insightful reviews! Model preference bias occurs at the distributional level, and the overall scoring behavior can be viewed as a mixture of category-specific distributions. While fine-grained, category-level calibration is a natural extension, at the cost of introducing additional complexity in labeling categories. Given that our main contribution is a simple and efficient calibration method, we originally used a single global offset in the main paper
>
> We find your suggestion helpful, and we conducted an additional experiment where we used GPT-5 to categorize the prompts in our calibration dataset into six types: **Math/Reasoning, Creative Writing, Code Generation, Factual QA, Instruction Following and Others**. For each category, we computed the Mismatch Degree and optimized the offset between Gemma-2-9b-it-SimPO and GPT-4o-mini using Skywork-RM. We then applied CHARM separately within each category and merged the resulting calibrated datasets into a single dataset for reward-model training.
>
> The category-wise calibration yields an additional improvement of 0.9 points on average RM-Bench, which is encouraging and also consistent with our expectations. This means that for scenarios where maximizing performance is the top priority, category-specific calibration can indeed provide further gains, and CHARM can naturally extend to this setting.
>
> Thanks again for the insightful suggestion. We will add a subsection to the paper to include this analysis and discuss the tradeoffs between global and category-wise calibration. Results shown in table re1 and re2.
>
> **Table Re1**: The Mismatch Degree for each category.
> | **Category**          | **MD** |
> |-----------------------|--------|
> | Code Generation       | 0.736  |
> | Creative Writing      | 0.538  |
> | Factual QA            | 0.894  |
> | Instruction Following | 0.411  |
> | Math/Reasoning        | 0.322  |
> | Others                | 0.875  |
>
> **Table Re2**: RM-Bench performance of different offset type calibration.
> | **Offset Type**     | **Chat** | **Math** | **Code** | **Safety** | **Avg** |
> |---------------------|:--------:|:--------:|:--------:|:----------:|:-------:|
> | Original Skywork-RM | 68.7     | 62.0     | 52.8     | 95.9       | 69.9    |
> | global offset       | 72.5     | 62.9     | 54.4     | 95.7       | 71.4    |
> | category offset     | 74.2     | 64.2     | 54.5     | 96.2       | 72.3    |
>
>
> > W2: Weak Ground Truth Assumption: The method treats Chatbot Arena Elo scores as absolute ground truth for human preferences, but these scores have well-documented limitations: they reflect a specific user population, are influenced by positional bias, and conflate multiple criteria (helpfulness, safety, style). The paper doesn't address how Chatbot Arena's biases might propagate into CHARM. For example, if Arena users prefer verbose responses, CHARM might inadvertently bake this length bias into the RM rather than remove it.
>
> We appreciate your concern and would like to clarify that CHARM uses Chatbot Arena Elo scores as a data source, for the purpose of demonstrating our algorithm, rather than building a bias-free reward model (esp. if we consider the data biased). We argue that this identified limitation does not undermine our approach. Technically, CHARM does not treat Arena Elo as "absolute ground truth" for response quality, but uses it to establish relative model rankings. Our calibration only requires that higher-Elo models should win more often, with win rates matching Elo predictions. CHARM can be run on any model ranking data; as you suggested earlier, CHARM can also be run on a subset of such data to get a more fine-grained calibration.

---

> ### Author Response · Authors · 2025-11-16
> **Response to Reviewer q9aZ (2/2)**
>
> > W3: Limited Calibration Scope: The main experiments calibrate using only one over-valued model (Gemma-2-9b-it-SimPO) and one reference model (GPT-4o-mini). This is a major limitation: (1) The paper claims bias is systematic across "preference-optimized" models, but only demonstrates calibration on a single instance. Testing with multiple over-valued models (e.g., DPO-tuned, PPO-tuned) is essential to validate generalizability. (2) The offset Δ is fundamentally tied to the choice of π_R. The paper fails to explore how Δ changes with different references (e.g., a weaker model like Llama-3-3.1-8B vs. GPT-4o). This raises questions about the stability and interpretability of the calibration.
>
> Re (1), most preference-optimized models listed in Appendix A.1 lack Elo scores (as they did not participate in Chatbot Arena at the time of our experiments). We are conducting additional experiments with some DPO-tuned models, and will report the results once the experiment is done.
>
> Re (2), Section 6.1 and Table 2 demonstrate generalization across multiple over-valued models with varying characteristics. We apologize for a labeling error in Table 2, the column header "**Ref Models**" should be "**Over-valued Models**" (the reference model was fixed as GPT-4o-mini). This table shows calibration results for 6 different over-valued models:
>
> - gemma-2-9b-it-SimPO (MD=0.639)
> - gemma-2-27b-it (MD=0.225)
> - gemma-2-9b-it (MD=0.155)
> - Qwen2.5-72B-Instruct (MD=0.088)
> - Llama-3.1-70B-Instruct (MD=0.048)
> - Llama-3.1-8B-Instruct (MD=0.032)
>
> We observed a strong correlation between Mismatch Degree and performance improvement, demonstrating that CHARM's effectiveness is systematic across different over-valued models.
>
> Also, we conducted new experiments to your suggestions, fixing the over-valued model (Gemma-2-9b-it-SimPO) and varying the reference model to explore how calibration performance change. Results see Table re3.
>
> Again, we see consistent Improvements as all reference model choices lead to performance gains over the original RM, demonstrating robustness. Also there exhibits a correlation between MD and performance gains. Reference models yielding higher MD values (e.g., Llama-3.1-70B: MD=0.675) tend to produce larger improvements, while lower MD values (e.g., Gemma-2-9b-it: MD=0.582) yield moderate gains. This is consistent with our MD-as-indicator finding (Section 6.1) and proves stability and interpretability.
>
> **Table Re3**: RM-Bench performance of different choice of reference models.
>
> | **Over-valued Model** | **Reference Model**    | **MD** | **Chat** | **Math** | **Code** | **Safety** | **Avg** |
> |-----------------------|------------------------|:------:|:--------:|:--------:|:--------:|:----------:|:-------:|
> | Original Skywork-RM   |                        | -      | 68.7     | 62.0     | 52.8     | 95.9       | 69.9    |
> | gemma-2-9b-it-SimPO   | GPT-4o-mini            | 0.639  | 73.9     | 62.4     | 53.9     | 95.8       | 71.5    |
> | gemma-2-9b-it-SimPO   | Gemma-2-9b-it          | 0.582  | 70.7     | 63.2     | 54.6     | 94.7       | 70.8    |
> | gemma-2-9b-it-SimPO   | Gemma-2-27b-it         | 0.633  | 70.0     | 63.1     | 54.0     | 94.7       | 70.4    |
> | gemma-2-9b-it-SimPO   | Llama-3.1-8B-Instruct  | 0.506  | 71.4     | 64.5     | 53.7     | 95.9       | 71.4    |
> | gemma-2-9b-it-SimPO   | Llama-3.1-70B-Instruct | 0.675  | 72.2     | 64.7     | 55.0     | 95.2       | 71.8    |
> | gemma-2-9b-it-SimPO   | Qwen2.5-72B-Instruct   | 0.625  | 70.9     | 62.4     | 53.8     | 96.0       | 70.8    |
>
> > Q1: What is the reward model in line 240 to produce the uncalibrated preference dataset? Will the choice of this reward model influence the final results?
>
> Thanks for your questions. We use the reward model being calibrated itself to construct the uncalibrated preference dataset. For example, when calibrating Skywork-RM, we use Skywork-RM to score responses from the over-valued and reference models, construct uncalibrated preference dataset based on these scores and then apply CHARM to compute offset Δ and create the calibrated dataset.
>
> This is a self-calibration process independently for each RM. Table 1 reports results of five different RMs being calibrated using this approach. The consistent performance gains across all RMs demonstrate the robustness of our method. Since each RM undergoes self-calibration, there is no "choice" in our experimental design. Using each RM's own scores ensures CHARM corrects that specific RM's systematic biases.
>
> We will make this point clearer in the revised manuscript.
>
> ---
>
> Thanks again for your careful review and insightful suggestions. We hope our responses adequately address your concerns and help clarify and strengthen our contributions. If you have any further questions or would like us to elaborate on any point, please feel free to let us know. We will do our best to respond.

---

### Author Response · Authors · 2025-11-23
**Manuscript revised. Looking forward to further discussion.**

We sincerely thank all the reviewers for their valuable suggestions on our work. We have updated the revised version of our manuscript and highlighted the modified parts. We summarize our modification as below:

1. **Related Work about Classifier Calibration**
    - **Lines**: 89–94
    - **Related Questions**: Reviewer Wq5j W4
    - **What we have done**: We expand the definition of classifier calibration and the related work, clarify how we generalize the concept of “classifier calibration” to reward model calibration, and more clearly explain the difference between sample-wise and model-wise calibration.
2. **Results on Model Post-training**
    - **Lines**: 241–246, 354–369, Table 2
    - **Related Questions**: Reviewer Wq5j W3
    - **What we have done**: We add more details on the DPO post-training experiments, including the data construction procedure and results compared with baseline and original RM.
3. **Clarification on RM Choice**
    - **Lines**: 253
    - **Related Questions**: Reviewer q9aZ Q1
    - **What we have done**: We clarify on the reward model choice when we scoring the uncalibrated preference datasets.
4. **Clarification on Model Pair Choice**
    - **Lines**: 265–269
    - **Related Questions**: Reviewer B6Db W2 Q2
    - **What we have done**: We explain why we choose the pair gemma-2-9b-it-SimPO vs. GPT-4o-mini and empirical evidence for using MD as the selection criterion.
5. **Results on CHARM Generalization**
    - **Lines: 418–421**, Table 3
    - **Related Questions**: Reviewer q9aZ W3, Reviewer Wq5j W2
    - **What we have done**: We add experimental analysis and explanations on different model pairs, emphasize that CHARM can generalize rather than only being effective for the chosen pair, and discuss whether model bias is related to the mismatch degree.
6. **Discussion on GRMs**
    - **Lines**: 509–512
    - **Related Questions**: Reviewer 8Hwd W2
    - **What we have done**: We add more discussion details on generative reward models, emphasize that model preference bias also exists in GRMs, and explain why GRM calibration is left as future work.
7. **Results on Category-wise CHARM**
    - **Lines**: 485–501, 951–967, Table 8
    - **Related Questions**: Reviewer q9aZ W1
    - **What we have done**: We add experiments on category-wise calibration; explain why category-wise CHARM is more complex but more effective; and clarify the trade-off between improvement and annotation cost.
8. **Robustness on Elo Updates**
    - **Lines**: 970–1005, Table 9
    - **Related Questions**: Reviewer B6Db W1 Q5
    - **What we have done**: We explain whether dynamic updates of Elo ratings affect calibration, include an extra experiment, and emphasize the robustness of CHARM to Elo updates.
9. **Results on Finetuning on Arena Battles**
    - **Lines**: 1010–1025, Table 10
    - **Related Questions**: Reviewer 8Hwd W1, Reviewer B6Db Q1
    - **What we have done**: We added a section explaining why we did not directly use Arena battles to finetune the reward model. Additionally, we conducted an extra experiment evaluating this baseline, showing that directly training on Arena battle data fails to mitigate model preference bias and even leads to performance degradation.
10. **Results on Multiple Model Calibration**
    - **Lines**: 1047–1074, Table 11
    - **Related Questions**: Reviewer Wq5j W2
    - **What we have done**: We investigate whether calibrating multiple models simultaneously works and extend CHARM to this secnario.
---
We are looking forward to your kind response. Please feel free to let us know any further questions. We would be happy to provide more clarification.

---

### Author Response · Authors · 2025-12-02
**TL;DR A Summary of Discussion by Authors**

Dear Reviewers, AC and Researchers,

Thank you for your time and effort, especially during a challenging period for the ICLR community. We sincerely appreciate the insightful feedback, which has helped us significantly improve our manuscript. In the revised PDF, we incorporated all additional experiments and clarifications conducted during the rebuttal period, addressing most of reviewers’ concerns. A complete list of modifications is provided in our official comments.

We first recap the major concerns from reviewers:

- **Method Generalization and Stability**, including model pair selection, robustness to Elo updates, category-wise calibration, multi-model calibration, and finetuning on Arena battle data. *(W1, W3 @ Reviewer q9aZ; W1, W2, Q1, Q2, Q5 @ Reviewer B6Db; W2 @ Reviewer Wq5j; W1 @ Reviewer 8Hwd)*
- **Method Application**, specifically results on downstream model post-training. *(W3 @ Reviewer Wq5j)*

Now we summarize our discussion as below:

**Reviewer q9aZ**  *score remained at 6; no follow-up discussion*

- **W1:** We added category-wise calibration experiments. Results (Tables 7 and 8) show additional performance gains from CHARM.
- **W2:** We clarified that CHARM uses Arena Elo only to establish relative model rankings, not as an absolute ground-truth signal.
- **W3:** We expanded experiments to include more diverse model pairs and demonstrated consistent performance improvements across them (Table 3).
- **Q1:** We clarified the reward model used to produce the uncalibrated preference datasets.

**Reviewer B6Db**  *score increased from 4 → 6 (Nov 29, 2025, 03:53 AOE)*

- **W1.1:** We clarified the calibration dataset curation pipeline and showed that CHARM is both scalable and efficient in practice.
- **W1.2 & Q5:** We investigated calibration under old vs. updated Elo ratings. Results confirm that recalibration is unnecessary (Table 9).
- **W2 & Q2:** We improved the explanation of our model-pair selection pipeline and clarified it in the revised manuscript.
- **Q1 & Q3:** We explained the data sparsity of Arena raw battles, and added an experiment showing that directly finetuning RMs on Arena battles leads to performance degradation.
- **Q4:** We clarified that training-free methods such as temperature scaling are compatible with CHARM.

**Reviewer 8Hwd**  *score remained at 4; follow-up question addressed, awaiting response*

- **W1:** We clarified the data sparsity of Arena battle data and provided experiments showing that RM finetuning on Arena battles degrades performance (Table 10).
- **W2:** This appears to be the reviewer’s primary reason for maintaining score. We respectfully disagree. CHARM targets scalar RMs because that is where we observe model preference bias. GRM inapplicability is shared by most RM calibration methods and does not diminish our contribution at all. We additionally conducted empirical analysis showing that model preference bias also exists in GRMs, and we provided insights on why mitigating this bias likely requires improving GRM reasoning ability rather than boundary calibration. We leave this direction for future work, as it is beyond the intended scope of CHARM and should not be considered a weakness of our method (Table 6).
- **W3:** We clarified that CHARM is unaffected by raw vs. style-controlled Elo, and provided experiments demonstrating this robustness (Table Re3).

**Reviewer Wq5j**  *score remained at 4; follow-up question addressed, awaiting response*

- **W1:** We clarified our rationale for treating Arena rankings as a strong proxy for correcting RM model preference bias.
- **W2:** We extended experiments with more diverse model pairs and showed consistent robustness. We additionally performed multi-model calibration, showing CHARM remains effective (Table 3).
- **W3:** This was the reviewer’s main concern. In the second rebuttal round, we added a DPO post-training experiment showing that CHARM-calibrated RMs lead to improved downstream policy model performance. As the reviewer noted, this result would significantly strengthen the contribution of our work. We believe Reviewer Wq5j would have considered raising the score had the rebuttal period not been shortened (Table Re3).
- **W4:** We clarified the distinction between calibration in supervised learning context and calibration in reward modeling. We also added a related work subsection in revised manuscript.
- **Q1:** We explained why head-to-head winrates are insufficient due to data sparsity, and why Elo-derived winrates provide a more stable and robust estimate.

We sincerely appreciate the reviewers’ and AC’s thoughtful feedback. The extensive experiments and clarifications we added directly address the core concerns and significantly strengthen the technical contribution of CHARM. We hope the improvements made during rebuttal sufficiently resolve any remaining doubts, and we look forward to your final assessment.

Best, Authors

---

### Meta-Review · Area_Chair_Y4ou · 2026-01-06

**Summary:**

1. The reliability of using LMSYS Elo scores as a ground truth. This may propagate existing biases in the LMSYS data.
2. Technical concerns about the method's reliance on a simple global offset and if it works for GRM.
3. Limited experiments on over-valued model. Plus some discussions about if the method is scalable and can generalize beyond the specific model pairs tested.

**Reviewer Concerns:**

Authors additional experiments on more model pairs, calibration methods, and different Elo versions help mitigate the second and third concerns.

However, the concern of whether to rely on Elo as the "ground-truth" remains a somewhat philosophical disagreement for some reviewers. Also, the limitation of whether this method can be applied to GRM and generalizes to more over-valued models remains.

**Reviewer Scores:**

Reviewer q9aZ: likely remain the same score.

Reviewer B6Db: likely remain the same score or slightly increase it.

Reviewer 8Hwd: remain the same score as the reviewer explicitly says the GRM concern is not addressed

Reviewer Wq5j: remain the same score as the reviewer explicitly says that

---

### Decision · Program_Chairs · 2026-01-26

Reject